# A transmission-virulence evolutionary trade-off explains attenuation of HIV-1 in Uganda

François Blanquart[1,2,3]*, Mary Kate Grabowski[4,5], Joshua Herbeck[6,7], Fred Nalugoda[8], David Serwadda[8,9], Michael A Eller[10,11], Merlin L Robb[10,11], Ronald Gray[4,5,8], Godfrey Kigozi[8], Oliver Laeyendecker[12,13], Katrina A Lythgoe[1,2,3,14], Gertrude Nakigozi[8], Thomas C Quinn[12,13], Steven J Reynolds[12,13], Maria J Wawer[4,5], Christophe Fraser[1,2,3,15]

[1]MRC Centre for Outbreak Analysis and Modelling, Imperial College London, London, United Kingdom; [2]Department of Infectious Disease Epidemiology, Imperial College London, London, United Kingdom; [3]School of Public Health, Imperial College London, London, United Kingdom; [4]Department of Epidemiology, Johns Hopkins University, Baltimore, United States; [5]Bloomberg School of Public Health, Johns Hopkins University, Baltimore, United States; [6]International Clinical Research Center, University of Washington, Seattle, United States; [7]Department of Global Health, University of Washington, Seattle, United States; [8]Rakai Health Sciences Program, Entebbe, Uganda; [9]School of Public Health, Makerere University, Kampala, Uganda; [10]U.S. Military HIV Research Program, Walter Reed Army Institute of Research, Silver Spring, United States; [11]Henry M. Jackson Foundation for the Advancement of Military Medicine, Bethesda, United States; [12]Laboratory of Immunoregulation, National Institute of Allergy and Infectious Diseases, National Institutes of Health, Bethesda, United States; [13]Division of Intramural Research, National Institute of Allergy and Infectious Diseases, National Institutes of Health, Bethesda, United States; [14]Department of Zoology, University of Oxford, Oxford, United Kingdom; [15]Big Data Institute, Li Ka Shing Centre for Health Information and Discovery, Nuffield Department of Medicine, University of Oxford, Oxford, United Kingdom

*For correspondence:
f.blanquart@imperial.ac.uk

Competing interests: The authors declare that no competing interests exist.

**Abstract** Evolutionary theory hypothesizes that intermediate virulence maximizes pathogen fitness as a result of a trade-off between virulence and transmission, but empirical evidence remains scarce. We bridge this gap using data from a large and long-standing HIV-1 prospective cohort, in Uganda. We use an epidemiological-evolutionary model parameterised with this data to derive evolutionary predictions based on analysis and detailed individual-based simulations. We robustly predict stabilising selection towards a low level of virulence, and rapid attenuation of the virus. Accordingly, set-point viral load, the most common measure of virulence, has declined in the last 20 years. Our model also predicts that subtype A is slowly outcompeting subtype D, with both subtypes becoming less virulent, as observed in the data. Reduction of set-point viral loads should have resulted in a 20% reduction in incidence, and a three years extension of untreated asymptomatic infection, increasing opportunities for timely treatment of infected individuals.

## Introduction

To spread, a pathogen must multiply within the host to ensure transmission, while simultaneously maintaining opportunities for transmission by avoiding host morbidity or death (*Anderson and May, 1982*; *Alizon et al., 2009*). This creates a trade-off between transmission and virulence. This hypothesis permeates theoretical work on the evolution of virulence, but empirical evidence remains scarce (*Dwyer et al., 1990*; *Mackinnon and Read, 1999*; *Fraser et al., 2007*; *de Roode and Yates, 2008*; *Alizon et al., 2009*; *Cressler et al., 2015*). In HIV-1 infection, set-point viral load (SPVL), the stable viral load in the asymptomatic phase of infection, is a viral trait which is both variable and heritable (*Hollingsworth et al., 2010*; *Fraser et al., 2014*; *Hodcroft et al., 2014*), and has an important impact on the transmission cycle of the pathogen. In untreated infection, higher SPVL translates into higher per-contact transmission rates but also faster disease progression to AIDS and death. From the perspective of the transmission cycle, this creates a trade-off, under which an intermediate SPVL value maximises opportunities for transmission (*Fraser et al., 2007*). Indeed the transmission potential of a parasite is the product of the transmission rate and the time during which the host is alive and can transmit. The latter is approximately the time to AIDS in HIV as host death occurs shortly after the onset of AIDS and sexual activity may be reduced in the AIDS phase because of AIDS-associated symptoms (*Hollingsworth et al., 2008*). The virulence-transmission trade-off in HIV is important for understanding pathogenesis and is a possible explanation for the significant changes in HIV virulence reported over the last decades in North America and Europe. There, SPVL increased at an estimated rate of 0.013 (*Herbeck et al., 2012*) and 0.020 $\log_{10}$ copies/mL/year (*Pantazis et al., 2014*) over the last 28 years. Since many persons at risk of infection do not routinely obtain HIV testing (*Paz-Bailey et al., 2013*), such changes may lead to more transmission and more newly diagnosed patients presenting with advanced infection, despite the widespread availability of antiretroviral therapy (ART).

The virulence-transmission trade-off is a promising hypothesis to explain changes in virulence of HIV, but this hypothesis and its predictions have so far been approached in a piecemeal manner, by combining data on infectiousness, AIDS-free survival and the dynamics of SPVL from very different cohorts (*Fraser et al., 2007*; *Herbeck et al., 2012*; *Pantazis et al., 2014*). Here we integrated extensive data from a single cohort in Uganda into an epidemiological-evolutionary model describing the transmission cycle of HIV. We then compared predictions on the evolution of SPVL evolution to the observed trends in SPVL in this cohort.

## Results

We focused on one of the longest established generalised HIV epidemics, in rural Uganda, and used data collected as part of the Rakai Community Cohort Study (RCCS), a large and long-standing population-based open cohort conducted by the Rakai Health Sciences Program (RHSP) in Rakai District. We combined data on transmission rates and survival to estimate the evolutionary optimal distribution of SPVL for the RCCS cohort, and then compared it to the dynamics of SPVL over time from 1995 to 2012. ART probably had little effect on the evolutionary dynamics of SPVL in Uganda because it only became available in 2004 and is initiated at relatively late stage infection (CD4 < 250 cells/mm$^3$ from 2004 to January 2011, and at < 350 cells/mm$^3$ from February, 2011 to the time of writing, August 2016).

As in other HIV epidemics, we found that SPVL is highly variable in this population, with values ranging from $10^2$ copies/mL to $10^7$ copies/mL. SPVL was calculated for 647 individuals who had a positive HIV serologic test within two study visits of their last negative test ('HIV incident cases', *Table 1*; median time between last negative visit and first positive visit is 1.25 years), and for 817 participants in a serodiscordant partnership ('serodiscordant couples', *Table 2*).

We analysed transmission in 817 serodiscordant couples, in which one partner was positive (index partner), while the other was initially negative and at risk of infection during follow-up. This analysis revealed that higher SPVL was associated with significantly increased rate of transmission. Transmission between partners was modelled as a Poisson process, in which the instantaneous transmission rate is constant (*Fraser et al., 2007*). We allowed the transmission rate to be a function of SPVL, $\beta(v)$. We estimated all parameters by maximum likelihood and compared different models based on Akaike Information Criterion (AIC) (Materials and methods, *Figure 1—figure supplement 1*). The

**Table 1.** Epidemiological and demographic characteristics of the HIV-1 incident cases in the Rakai cohort, used for the analysis of time trends in SPVL and for the analysis of time to AIDS. *Multiple subtypes (possibly dual infection) ** Recombinants, primarily A/D.

| Gender | N | Mean SPVL, [0.025; 0.975] quantiles |
|---|---|---|
| F | 362 | 4.3 [2.3; 5.85] |
| M | 285 | 4.54 [2.3; 6.03] |
| Date of infection | | |
| 1995–1999 | 269 | 4.47 [2.3; 6.01] |
| 2000–2004 | 297 | 4.46 [2.3; 5.83] |
| 2005–2009 | 54 | 3.98 [2.3; 5.77] |
| ≥2010 | 27 | 3.97 [2.2; 5.33] |
| HIV-1 subtype | | |
| A | 96 | 4.34 [2.78; 5.61] |
| C | 6 | 3.92 [3.42; 4.71] |
| D | 292 | 4.56 [2.62; 5.92] |
| M* | 14 | 3.99 [2.48; 5.35] |
| R** | 74 | 4.38 [2.33; 5.84] |
| Unknown | 165 | 4.22 [2.3; 6.03] |
| Age at infection | | |
| 15–19 | 61 | 4.17 [2.3; 5.51] |
| 20–29 | 327 | 4.43 [2.3; 5.97] |
| 30–39 | 182 | 4.45 [2.3; 5.88] |
| 40–49 | 67 | 4.43 [2.28; 6.09] |
| ≥50 | 10 | 4.05 [2.3; 5.94] |

best model fit was one where transmission rates increases from 0.019/year to 0.14/year in a stepwise fashion as SPVL increases with three plateaus (*Figure 1a*) (ΔAIC = −75.96 compared to null model

**Table 2.** Epidemiological and demographic characteristics of the infected individual in serodiscordant couples in the Rakai cohort, used for the analysis of time trends in SPVL and for the analysis of time to AIDS. ** Including recombinants, primarily A/D.

| Gender | N | Mean SPVL, [0.025; 0.975] quantiles |
|---|---|---|
| F | 324 | 3.99 [2.3; 5.61] |
| M | 493 | 4.23 [2.3; 5.85] |
| Date of infection | | |
| Unknown | 595 | 4.1 [2.3; 5.64] |
| 1995–1999 | 93 | 4.13 [2.3; 5.53] |
| 2000–2004 | 96 | 4.41 [2.3; 5.98] |
| 2005–2009 | 30 | 4.08 [2.3; 5.62] |
| ≥2010 | 3 | 3.19 [2.36; 3.77] |
| HIV-1 subtype | | |
| A | 54 | 4.11 [2.42; 5.72] |
| D | 430 | 4.27 [2.4; 5.77] |
| Other/Unknown** | 333 | 3.97 [2.3; 5.67] |

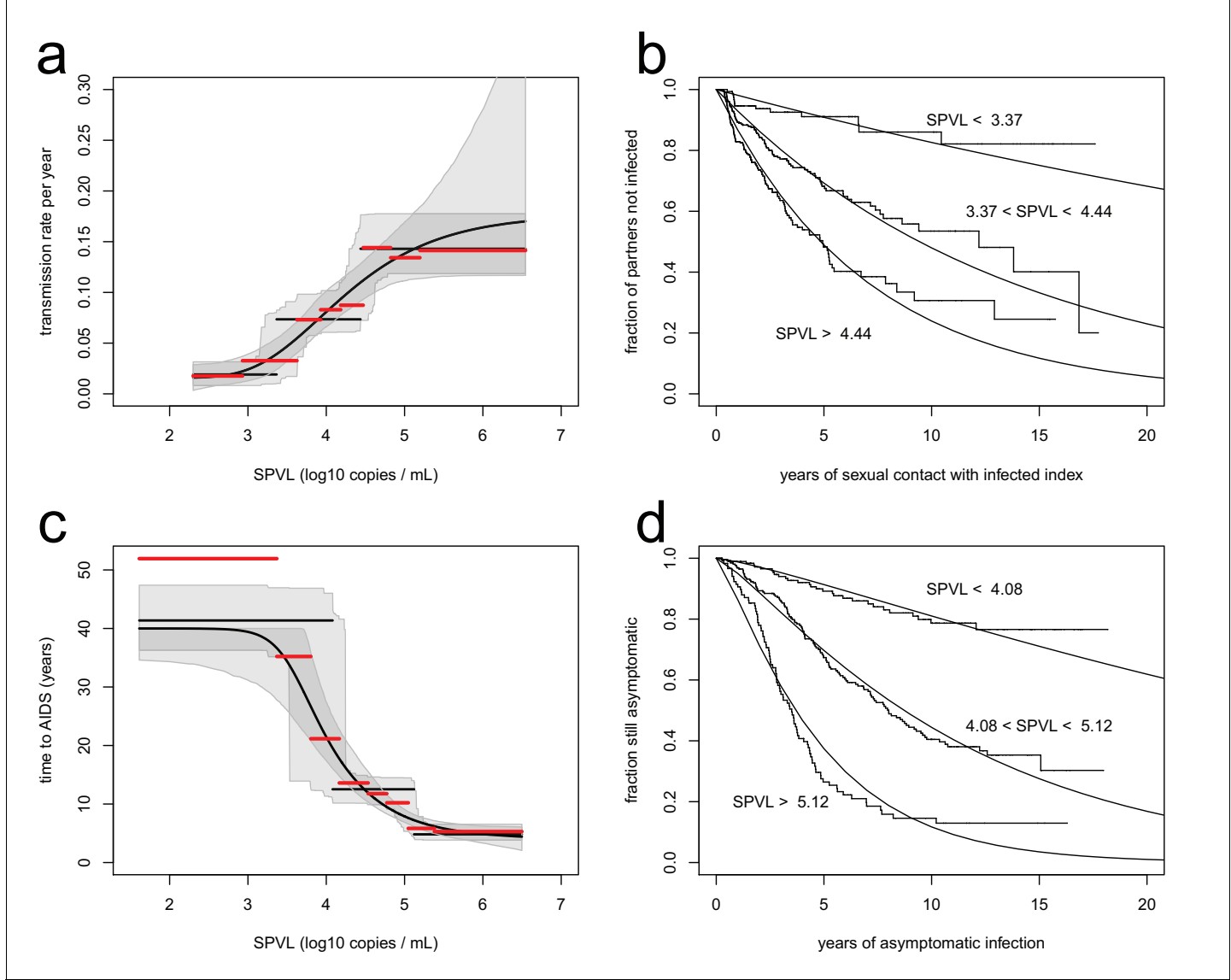

**Figure 1.** Inferred relationships between SPVL and transmission rate (**a**, **b**) and time to AIDS (**b**, **d**). On the left panels, black lines show the maximum likelihood relationships and shaded areas the bootstrap 95% confidence intervals. Both the step function (horizontal lines) and the generalised Hill function (curved line) are shown. The red lines show a non-parametric estimation of the transmission rate (**a**) and the time to AIDS (**c**) curves, when the data is stratified by SPVL in 8 bins of equal size. The right panels show Kaplan Meier plots when the data is partitioned in three SPVL groups defined by the maximum likelihood relationships. There was good agreement between the data (step functions) and the maximum likelihood function (smooth functions).

The following source data and figure supplements are available for figure 1:

**Source data 1.** Data file for *Figure 1*.

**Figure supplement 1.** Functional forms for time to AIDS (**a**), and transmission rate (**b**), as a function of SPVL..

**Figure supplement 2.** The inferred transmission rate (**a**) and time to AIDS (**b**), as a function of SPVL, are similar when removing undetectable SPVL values from the analysis.

**Figure supplement 3.** Transmission rate as a function of SPVL, stratified by gender (**a**) and by circumcision status (**b**, **c**).

with a fixed transmission rate, n = 817). A function with three steps was favoured over others, but we also show a continuous function, the generalised Hill function, that may be considered more biologically realistic (ΔAIC = - 71.17 compared to the null model, n = 817) (*Figure 1a*). The two functions fitted the data well, as shown by comparison with non-parametric estimates of the transmission rate in the data stratified by SPVL (*Figure 1a*), and by a Kaplan-Meier plot comparing data to the model prediction (*Figure 1b*). We also allowed the parameters of the function $\beta(v)$ to vary with the covariates subtype, gender, and male circumcision status. In accordance with previous studies (*Kiwanuka et al., 2009*), subtype A had a higher transmission rate than subtype D for all SPVL values (Figure 3) (ΔAIC = −3.32 compared to the model without subtype, n = 817). We will examine the evolutionary consequences of subtype differences later on. Gender did not have an effect on transmission (ΔAIC = 1.66 compared to model without gender, n = 817), and male circumcision reduced transmission both from female to male and from male to female (ΔAIC = −3.74 for female to male, n = 321; ΔAIC = −3.17 for male to female, n = 487, compared to model without circumcision) (*Figure 1—figure supplement 3*).

We assessed the relationship between SPVL and time to AIDS from 562 incident cases with a SPVL value and information on time to AIDS, and found that higher SPVL was associated with significantly shorter time to AIDS (*Figure 1c*). The time to AIDS was assumed to follow a gamma distribution, where the expected value was a function of SPVL (*Fraser et al., 2007*). We optimized the likelihood function and compared different models for the dependence of time to AIDS on SPVL based on AIC. The best model was a step function with three plateaus, with time to AIDS decreasing from 40 years to 5 years from low to high SPVL (*Figure 1c*; ΔAIC = 137.22 compared to null model with fixed time to AIDS). Again, non-parametric estimation of the time to AIDS (*Figure 1c*) and a Kaplan-Meier survival plot (*Figure 1d*) showed good fit of the model to the data. We also allowed the relationship between SPVL and time to AIDS to vary by subtype and gender. The inferred gamma distribution had shape parameter 1.2, similar to an exponential distribution (which is the special case where shape parameter is 1). We found, in agreement with previous studies (*Kiwanuka et al., 2008*), that subtype D tended to confer faster disease progression, but this effect was not statistically significant here (*Figure 1*, ΔAIC = 15.41 compared to the model without subtype, n = 562). However, subtype D-infected individuals who progressed rapidly were not included in the analysis because they had no SPVL value (among the 33 individuals who progressed to AIDS within 10 years but had no SPVL value, there were 12 subtype D, 1 recombinant, and 20 unknown subtype). Time to AIDS did not significantly vary by gender (*Figure 1*, ΔAIC = 7.85 compared to the model without gender, n = 562).

Next, we predicted how SPVL might change over time under the trade-off between virulence and transmission, incorporating our setting-specific estimates of the virulence-transmission trade-off into an evolutionary and epidemiological model. The model is an analytically tractable Susceptible-Infected compartmental ordinary differential equation (ODE) model, where the viral population is stratified by SPVL, similar to previous models of virulence evolution (*Day and Proulx, 2004*; *Day and Gandon, 2007*) (Material and methods). SPVL of an infected individual is the sum of a viral genetic effect *g*, which is transmitted with mutation from a donor to a recipient, and an environmental effect *e*, which includes host and other environmental factors and is independently drawn in a normal distribution with mean 0 in each newly infected individual. The evolution of mean SPVL in the population is determined by the evolution of the mean viral genetic effect *g*. In this model the transmission rate of a virus with SPVL *v* is the inferred function $\beta(v)$ (*Figure 1a*), while death is assumed to occur at a constant rate $\mu(v)$ given by the inverse of the mean time to AIDS (*Figure 1c*). In the ODE model, the time to AIDS follows an exponential distribution because the rate of AIDS-death is constant. The individual based model presented later on relaxes this assumption and considers gamma-distributed time to AIDS as inferred from the data.

We first developed an analytical expression for the evolution of SPVL. Because prevalence of HIV in this cohort is approximately constant (at 14% on average in the period 1995 to 2013, *Figure 2—figure supplement 1*) and the distribution of SPVL can be closely approximated by a normal distribution, we were able to use an approximation of the Price equation (*Price, 1970*) inspired by a classical quantitative genetics model (*Lande, 1976*), to write the change in mean genetic effect of SPVL in prevalent cases over time as (Appendix):

$$\frac{d\bar{g}}{dt} = \underbrace{V_P \, h^2 \, \frac{\bar{\mu}^2}{\bar{\beta}} \frac{\partial(\bar{\beta}/\bar{\mu})}{\partial \bar{g}}}_{transmission-virulence\,trade-off} + \underbrace{\alpha \bar{\mu}}_{within-host\,evolution}$$

The equation has two terms that respectively describe the effects of selection and of inheritance on SPVL evolution. The first term describes selection under a virulence-transmission trade-off, maximising the ratio of the mean transmission rate over the mean severity of infection, $\bar{\beta}/\bar{\mu}$, which is the mean fitness of the viral population. The SPVL that maximises mean fitness is 3.4 $\log_{10}$ mL/copies (95% bootstrap CI [2.6; 4.0], *Figure 2a*). Adaptation of the viral population will proceed at a rate proportional to phenotypic variance $V_P$ (the variance in SPVL) and heritability $h^2$ (the fraction of variance explained by viral genetic factors, assumed to be at equilibrium). The second term describes biased mutation that changes the mean SPVL from one infection to the next, where $\alpha$ is the mean effect of mutations from the donor to the recipient, recapitulating the effect of within-host selection on mean SPVL. The effects of the transmission-virulence trade-off were very similar when we used the generalised Hill functional form to fit the relationships between SPVL and transmission and time to AIDS (*Figure 2a*).

Next we simulated the ODE and assessed the precision of the analytical approximation. We parameterised the ODE model with the data and simulated the evolution of mean SPVL from 1995 to 2015. Parameterisation was as follows: the transmission rate was as in *Figure 1a*; the mortality rate was the inverse of time to AIDS (*Figure 1c*); heritability of SPVL in the Rakai cohort was previously estimated at 36% (confidence interval 6–66%), using 97 donor-recipient transmission pairs (*Hollingsworth et al., 2010*) (who are participants of the present cohort). We had little data to parameterise the effect of within-host evolution on SPVL, $\alpha$. Many different types of mutations may evolve within the host, and little is known on the net effect of these processes on SPVL. Within-host viral fitness is positively related to replicative capacity (RC), measured in the absence of an immune response, and immune escape, which is host-specific. Most studies of within-host HIV evolution have focused on CTL escape mutations, which are conditionally beneficial (i.e. their positive effect on fitness is host-specific). These usually sweep through during infection because the fitness benefit of evading the immune system outweighs the cost of reduced RC that these mutations also impose (*Goepfert et al., 2008*; *Carlson and Brumme, 2008*; *Matthews et al., 2008*). CTL escape mutations may be reverted if the virus harbouring a costly CTL-escape mutation is transmitted to an individual where the mutation does not help evade the new host's immune system (*Carlson et al., 2014*; *Zanini et al., 2015*). Mutations that increase the replicative capacity of the virus in all hosts may also evolve (*Kouyos et al., 2011*). It is also a possibility that slightly deleterious or beneficial mutations get fixed by genetic drift. We explored three scenarios where available data allow rough estimation of plausible values for the impact of within-host evolution on viral load (the $\alpha$ parameter) (Material and methods). (i) Most mutations evolving are conditionally beneficial but carry a strong cost to RC ($\alpha = -0.47$ $\log_{10}$ copies/mL). (ii) Most mutations evolving are conditionally beneficial but carry a moderate cost to RC ($\alpha = -0.093$ $\log_{10}$ copies/mL). (iii) Most mutations have unconditionally beneficial effects on RC ($\alpha = +0.057$ $\log_{10}$ copies/mL).

The ODE simulations predicted a decline in mean SPVL in incident cases from 1995 to 2015, at a rate of $-0.042$, $-0.013$ and $-0.0009$ $\log_{10}$ copies/mL/year in the three scenarios chosen for within-host evolution, for a heritability of 36%. The Price equation predicted the outcome of the ODE simulations quite precisely (*Figure 2b*). The Price equation shows that the virulence-transmission trade-off – the first term in the equation – contributes initially a decline in mean SPVL of $-0.01$ $\log_{10}$ copies/mL/year, slowing down as the population gets closer to the optimum. Note that the Price equation concerns average genetic effect of SPVL in the *prevalent* cases, but the rate of evolution in the *incident* cases was similar in these simulations (*Figure 2—figure supplement 6*). Predictions of the ODE model were robust to the addition of a number of more realistic features of the HIV epidemic, as shown by a more comprehensive individual-based stochastic model (IBM) of HIV evolution (*Herbeck et al., 2014, 2016*). The IBM includes all features of the ODE model, in particular the fact that SPVL is the addition of a heritable genetic component and a random environmental component. In addition, it includes the phases of acute infection and AIDS, both characterized by viral loads being much higher than the set-point value. Disease progression was modelled as progression through a series of CD4 count categories until AIDS occurred, and the transition rates between these

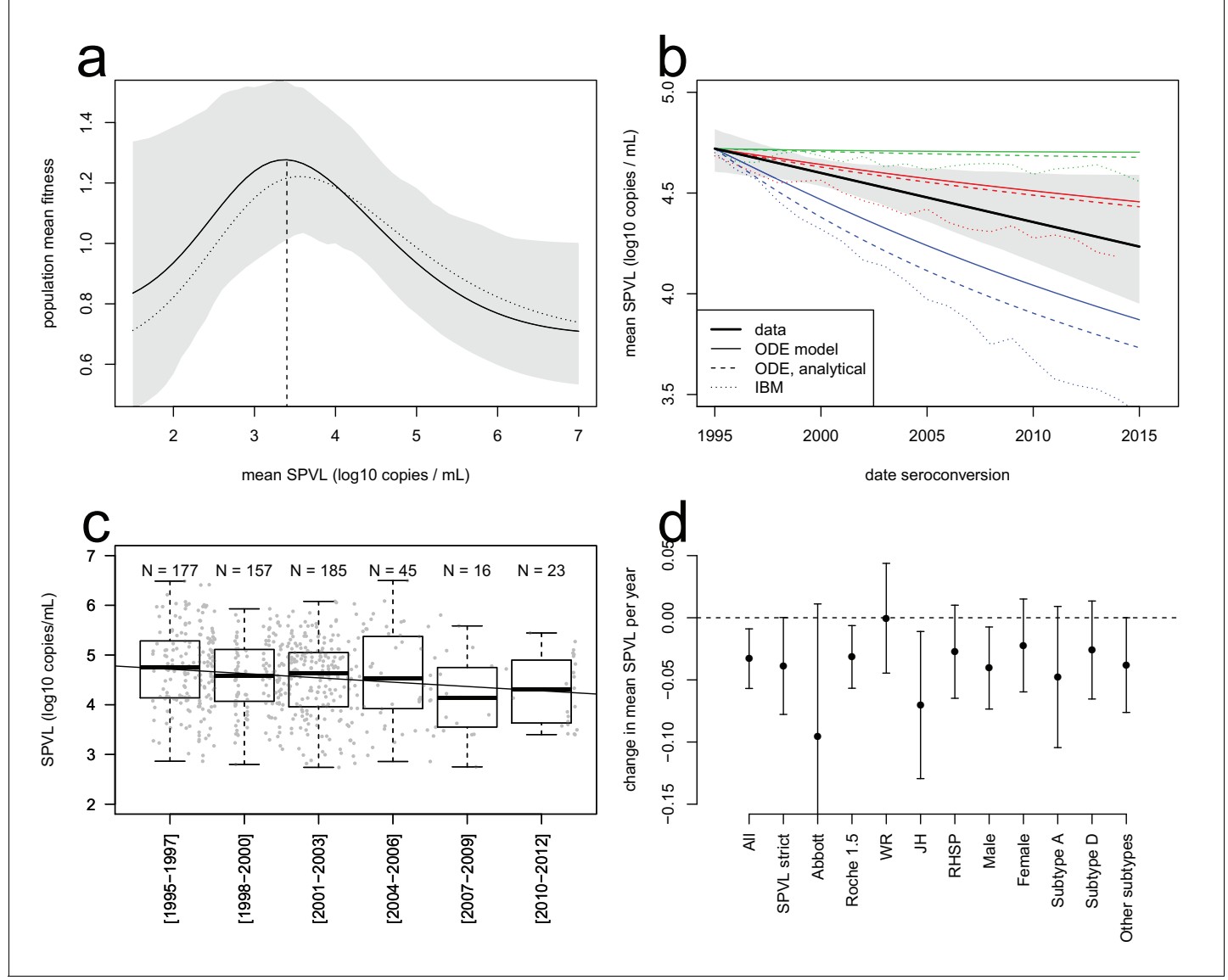

**Figure 2.** Evolutionary dynamics of SPVL. (**a**), mean fitness of the viral population as a function of mean SPVL when transmission and time to AIDS are fitted as step functions (solid line; shaded area shows the 95% C.I.) or generalised Hill functions (dashed line). (**b**), evolutionary predictions for the temporal dynamics of mean SPVL given by the ODE model (thin solid and dashed lines), and the stochastic IBM (dotted lines), under three scenarios for the impact of within-host evolution (biased mutation) on SPVL in blue (1, $\alpha = -0.47$ $\log_{10}$ copies/mL), red (2, $\alpha = -0.093$ $\log_{10}$ copies/mL) and green (3, $\alpha = +0.057$ $\log_{10}$ copies/mL). The thick line is the data, showing the linear regression of SPVL on date of seroconversion, with 95% bootstrap confidence intervals shown as a shaded area. (**c**), distribution of SPVL in the population over time; grey points show the data, and the line is the unadjusted regression of SPVL over time. (**d**) coefficient of regression of SPVL over time in the adjusted linear regression, with confidence intervals, in various subsets of the data (Material and methods). All data; SPVL strict definition; SPVL measured with Abbott assay and Roche 1.5 assay; SPVL measured at Walter Reed (WR), John Hopkins (JH) and RHSP laboratories; SPVL in males and females; subtype A, subtype D, and other/unknown subtype viruses.

The following source data and figure supplements are available for figure 2:

**Source data 1.** Data file for *Figure 2*.

**Figure supplement 1.** Prevalence of HIV over time, in the Rakai communities (gray lines), and on average across all communities (thick black line).

**Figure supplement 2.** Summary of effects for the multivariate linear model explaining SPVL (*Figure 2—source data 1*).

**Figure supplement 3.** ART had little impact on the evolution of SPVL under the virulence-transmission trade-off.

*Figure 2 continued on next page*

*Figure 2 continued*

**Figure supplement 4.** The entire distribution of SPVL shifts downwards with time.
**Figure supplement 5.** Declining prevalence had little impact on the evolution of SPVL under the virulence-transmission trade-off.
**Figure supplement 6.** Comparison of SPVL trends in incident cases and prevalent cases.

categories were tuned to reproduce the inferred gamma-distributed time to AIDS. Partnership formation and dissolution was also explicitly modelled, as well as some degree of behavioural heterogeneity in partnership duration and coital frequency. The IBM also predicted a decline in mean SPVL in the three scenarios, although at a somewhat faster rate compared to the simplified ODE model, confirming the generality and robustness of our results (*Figure 2b*).

Strikingly, the data was in qualitative agreement with the evolutionary model: SPVL in the Rakai cohort decreased with date of seroconversion between 1995 and 2012, at a rate of $-0.022$ $\log_{10}$ copies/mL per year after adjustment for other covariates (CI [$-0.04$; $-0.002$], p=0.027, n = 603) (*Figure 2*). Average SPVL in prevalent cases was also declining at a rate of $-0.020$ $\log_{10}$ copies/mL, although for those it is more difficult to adjust for covariates and test for significance (because the same participants are 'prevalent cases' at multiple time points) (*Figure 2—figure supplement 6*). The observed trends were best explained if mutations evolving within the host had a moderate negative impact on mean SPVL (scenario 2).

The agreement between the observed trend in mean SPVL and the evolutionary model suggests that genetic changes in the virus may be responsible for decreasing SPVLs. However, it is possible that other confounding effects might explain some or all of the decrease in SPVL. Because the Rakai cohort has been studied extensively, we were able to consider the potential impact of a number of confounders but none of them could explain the observed decline in mean SPVL of around 0.4 $\log_{10}$ copies/mL over 17 years (*Figure 2*). SPVL decline was significant in the linear model both without adjustment ($-0.029$ $\log_{10}$ copies/mL per year, CI [$-0.045$; $-0.013$], p=0.0005, n = 603, *Figure 2c*), and in the multivariate regression mentioned above, controlling for the laboratory where SPVL was measured, assay type, gender, age and subtype. Additionally, to verify the robustness of the decline in mean SPVL, we examined the trend in SPVL in a number of subsets of the population (*Figure 2d*). SPVL declined in a similar way: (i) when using the 'strict' definition of SPVL (i.e. the subset of measures that included more than one viral load measurement and where the standard error across viral loads of the same participant was less than one $\log_{10}$ copies/mL) (Appendix); (ii) within each gender (*Figure 2d*); (iii) within each assay type, when partitioning the data in viral loads measured with the 'Abbott' assays and the 'Roche 1.5' assays, showing that declining SPVL was not due to changing assays; (iv) for viral loads measured at the John Hopkins and at the RHSP laboratories; and it is unlikely there were independent downward shifts in assay reading over time in these two laboratories. Mean SPVL did not decline in the subset of SPVL measured in the Walter Reed laboratory, but 90% of those were for participants infected prior to 2003, limiting power to detect temporal trends.

Improvement in nutrition or health care could be hypothesised to cause a decline in SPVL over time. However, improvement in nutrition would probably have no impact on the mean SPVL, as improving micronutrient intake slows down disease progression, but does not reduce plasma viral load (*Fawzi et al., 2005*; *Friis, 2006*; *Baum et al., 2013*). According to a survey conducted in 2006 in the Rakai communities, households experience on average 2 months per year of food insecurity, and the Household Dietary Diversity Score is 7.7 / 12 (S. Haberlen, personal communication, August 2016), which is high enough to meet WHO dietary requirements in energy, proteins, minerals and vitamins (*Steyn et al., 2006*). Improved healthcare is also a possible confounder. ART was introduced in Uganda in 2004, but until 2011 ART was prescribed only at late stage infection (CD4 count below 250 cells/mL). Although we excluded post-ART viral load measures from SPVL calculations, unreported ART use could have become more frequent at later time points and therefore might have contributed the decline in mean SPVL. To exclude this possibility, we first verified that the entire distribution of SPVL shifted downward, and the decline in mean SVPL was not only due to more low viral loads at later time points (*Figure 2—figure supplement 4*). We also examined the individual viral load trajectories within participants to verify that the clear drop in viraemia caused by

ART was not present in more recent participants without reported ART (*Supplementary file 1*). Last we examined the determinants of SPVL using the same linear model, focussing on the subset of SPVL values with viral loads measured before 2004, prior to ART availability in the region. We found a similar though non-significant linear decline in SPVL after non-significant 'laboratory' factors were removed (effect size = −0.019 $\log_{10}$ copies/mL, CI [−0.052; 0.014], p=0.26, n = 442). In this subset of data, all SPVL but one were measured with the Roche 1.5 assay. We had little power to distinguish between 'laboratory' and 'calendar time' effects because of a strong correlation between these factors (ΔAIC = −1.9 for a model with "laboratory" relative to a model with 'calendar time'). However we know from the analysis of the full dataset that 'laboratory' has no significant effect on SPVL, and furthermore the inferred effects of 'laboratory' in the pre-2004 subset are consistent with confounding by calendar time and different from those of the full dataset, which suggests the temporal effect is the genuine effect here.

Coinfections such as tuberculosis, malaria, the herpes simplex virus 2, gonorrhea, or syphilis, might increase viral load in HIV infected individuals (*Modjarrad and Vermund, 2010*). Better health care in the Rakai district could have caused a population-level reduction in SPVL via a reduction in prevalence of these coinfections. However, none of these coinfections had a combination of high prevalence at the beginning of the study, a large reduction in prevalence between 1995 and 2012, and a large effect on SPVL, sufficient to explain a decline of 0.4 $\log_{10}$ copies/mL (Material and methods).

To corroborate the evolutionary model, we extended it to include data on the subtype-specific transmission rate and model jointly the evolution of SPVL and subtype A, D, and AD recombinants (the major subtypes circulating in the population). The evolutionary model predicted the observed dynamics of subtype A, D, and AD recombinants ('R') in the cohort (*Figure 3*). In particular, HIV subtype A was more transmissible than subtype D for a given SPVL (*Kiwanuka et al., 2009*), and therefore was predicted to increase in frequency in the population. Temporal trends in subtype frequency in the data were inferred by focusing on subtypes A, D, and R and fitting a multinomial linear model for the frequency of the three subtypes as a function of seroconversion date. This revealed significant changes in subtype frequencies (analysis of deviance, p=0.044, n = 551) an increase in the frequency of subtype A (0.009 per year, bootstrap CI [−0.0007; 0.022]) and recombinants (0.007 per year, CI [−0.005; 0.017]), and a decrease in subtype D (−0.016, CI [−0.027; −0.002]), in accordance with a previous study (*Conroy et al., 2010*). The rise of subtype A and R together with the lower SPVL associated with infection with these subtypes contributes additionally to the decline in mean SPVL, but this effect is estimated at −0.003 $\log_{10}$ copies/mL/year, very small compared to the within-subtype evolution of SPVL at a rate of −0.022 $\log_{10}$ copies/mL/year (Material and methods). To model the dynamics of subtype A, D, and R within the ODE model, we assumed co-infection by A and D occurred only transiently and resulted in an 'R' infection with probability *r* (*Day and Gandon 2012*). We assumed the transmission function for subtype R was intermediate between that of subtype A and subtype D. In spite of large uncertainty in the fitness function of subtype A due to smaller numbers of infected individuals (*Figure 3c*), the model accurately predicted the rise in frequency of both subtypes A and R for *r*=1 (*Figure 3d*). SPVL declined within subtype A and D, the two major subtypes co-circulating in the region (*Figure 2d*). The inferred fitness functions for subtype A and D were both consistent with a decline in SPVL within each subtype (*Figure 3e*). We note, though, that the model predicted a slower decline in SPVL within subtype A than the one observed, because this subtype is expanding in the population, which favours selection for transmission and slows down the attenuation of the virus.

## Discussion

Using extensive data on a population-based cohort in the Rakai district, Uganda, we confirmed the existence of a virulence-transmission trade-off in HIV, and predicted that the viral population should evolve reduced SPVL to maximise transmission opportunities. This prediction was verified, as mean SPVL in newly infected participants declined by 0.4 $\log_{10}$ copies/mL in the Rakai cohort form 1995 to 2012. We had limited information on the impact of within-host evolution on mean SPVL. However, the virulence-transmission trade-off was not negligible compared to the potential impact of within-host evolution, and results in a decline in mean SPVL of −0.01 $\log_{10}$ copies/mL/year, i.e. about 50% of the observed trend. We systematically examined potential confounders in this well-studied cohort,

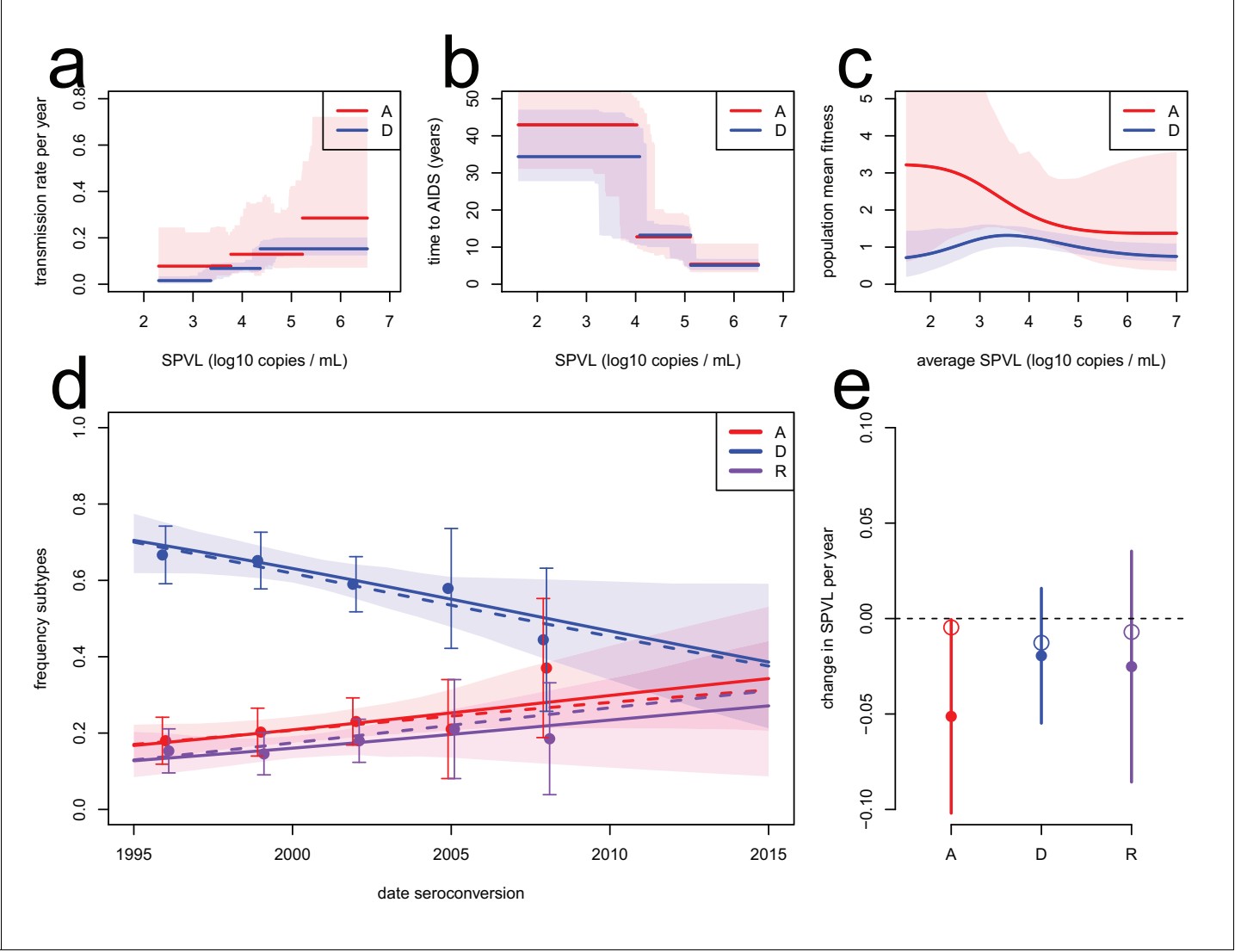

**Figure 3.** Subtype-specific evolutionary predictions. Maximum likelihood functions for transmission (**a**) and time to AIDS (**b**) as a function of SPVL, stratified by subtype, for heritability $h^2=0.36$ and biased mutation $\alpha = -0.093$ (scenario 2). Shaded areas are bootstrap confidence intervals. (**c**) Predicted fitness function for subtype A (red) and subtype D (blue). (**d**) Subtype dynamics in the Rakai cohort as inferred by fitting a multinomial linear model with a 'date seroconversion' effect (solid lines, and confidence intervals as a shaded area; points show the actual frequency in the data, binned in five time categories, with confidence intervals), together with subtype dynamics predicted by the ODE model stratified by subtype (dashed lines). Recombination occurs upon co-infection and generates 'R' subtypes (purple). (**e**) Rates of evolution of SPVL per year within subtype, in the data (points, with 95% confidence intervals) and in the ODE simulation stratified by subtype (open circles).

but none of them could account for the trend of declining SPVL, suggesting viral genetic changes may be responsible for the observed attenuation. The evolutionary model also quantitatively reproduced how higher transmission of subtype A resulted in expansion of this subtype in the population.

The attenuation of HIV in this Ugandan cohort is in contrast to increasing virulence in Europe. The European dynamics were hypothesized to result from viral adaptation to a higher optimal SPVL of 4.5 $\log_{10}$ copies/mL (*Fraser et al., 2007*; *Herbeck et al., 2014*). However this higher optimum was computed using a Zambian cohort for transmission estimates, and a Dutch cohort for time to AIDS (*Figure 1—figure supplement 1*). Transient selection for increased virulence could also have been important in Europe, and in fact SPVL has declined since 2004 (*Pantazis et al., 2014*). Our finding of HIV attenuation is consistent with another study of the evolution of HIV virulence in Africa.Comparison between the epidemic in Botswana and the younger epidemic in South Africa revealed declines

in SPVL, which was hypothesized to be due to the fixation of mutations conferring adaptation to HLA variants and decreased replicative capacity (*Payne et al., 2014*).

Although the agreement between the observed trend in mean SPVL and the evolutionary model are consistent with genetic changes in the virus causing decreasing SPVLs, genomic data is lacking to positively demonstrate viral genetic changes. Even if genomic data were available, this would be a challenging task since SPVL is probably determined by many loci of small effect (*Bartha et al., 2013*), and polygenic adaption is difficult to detect (*Pritchard et al., 2010*). However, adaptation of the viral population to the low optimum is a logical consequence of the impact of SPVL on transmission and time to AIDS, two robust relationships inferred from the data (*Figure 1*). These effects of SPVL on the viral transmission cycle, together with 30–40% viral heritability of SPVL (36% specifically in the Rakai cohort, but generally around 30–40% in different settings, [*Fraser et al., 2014*; *Mitov and Stadler, 2016*; *Leventhal and Bonhoeffer, 2016*]), is predicted to result in attenuation of the virus.

The detailed evolutionary model of HIV SPVL evolution presented here quantitatively reproduced the attenuation of HIV-1 virulence that happened in the last 20 years. This decline in virulence is predicted to continue into the future. This decline is unaffected by ART becoming more widely available, as even aggressive test-and-treat strategies have little predicted effect on these evolutionary dynamics (*Roberts et al., 2015*; *Herbeck et al., 2016*) (*Figure 2—figure supplement 3*). As ART becomes more widely available, essentially shortening the duration of infection, reduced SPVL will contribute to reductions in onwards transmission, and so synergise with efforts to eliminate the pathogen.

## Materials and methods

The RCCS has conducted regular surveys (approximately annual) of all consenting residents aged 15–49 in the same 50 communities since 1994, collecting detailed information on demographics, sexual behaviours and health status and obtaining blood for HIV testing from all consenting participants. Personal information on marital and long-term consensual partners is also collected, which enables retrospective identification of stable couples. All individuals found to be HIV-infected are referred for care, including CD4 T cells count and viral load measurements. Virtually all HIV transmission in this population is via heterosexual vaginal intercourse, and the rates of reported intercourse per week and month were found to be stable by HIV subtype and different study time periods.

### SPVL

SPVL was calculated for 817 participants in a serodiscordant partnership ('Serodiscordant couples', *Table 2*), and for 647 individuals who had a positive HIV serology test within two study visits of their last negative test ('HIV incident cases', *Table 1*; median time between last negative visit and first positive visit is 1.25 years). SPVL was defined as the mean $\log_{10}$ viral load for all visits occurring more than 6 months after estimated date of infection and before initiation of ART. Clinical records indicating ART initiation were available for participants who received care at an RHSP clinic prior to 2013. After 2013, ART care at most RHSP clinics was transferred to the Ugandan Ministry of Health. We determined receipt of treatment from clinics other than RHSP prior to 2013, or at any clinic post-2013, by self-reported ART treatment status (SI).

### Transmission

Transmission was modelled as a Poisson process, in which the instantaneous transmission rate is constant (*Fraser et al., 2007*). We allowed the transmission rate to be a function of SPVL and other epidemiological covariates. For a seropositive individual (the 'index') with SPVL $v$, the probability that infection of the seronegative partner occurs between time $t_{p,-}$ and $t_{p,+}$ (where the subscript $p$ stands for partner) is given by:

$$P\left[t_{p,-} < t_p^* < t_{p,+}\right] = e^{-\beta(v)\left(t_{p,-} - t_{init}\right)} - e^{-\beta(v)\left(t_{p,+} - t_{init}\right)}$$

where $t_{init}$ is the time at which the index becomes infected (defined as the mid-point between last negative and first positive dates) or where observation of the couple starts, whichever occurs last and $\beta(v)$ is the transmission hazard. In a Poisson process, the time to transmission is exponentially

distributed: thus the probability is obtained by integration of the probability density function of the exponential distribution between time $t_{p,-}$ and $t_{p,+}$. When infection occurred within the window of observation, $t_{p,-}$ and $t_{p,+}$ are simply the last time the partner was seen negative and the first time he/she was seen positive. When infection did not occur within the window of observation, $t_{p,-}$ is the last time the partner was seen and $t_{p,+}$ is infinity. The likelihood function is the product of these probabilities over all couples. We compared several functional forms for $\beta(v)$, including a flat function where viral load has no impact on transmission, a power function $\beta(v) = \beta_0 10^{kv}$, the Hill function $\beta(v) = \beta_{max} \frac{1}{1+10^{-k(v-v_{50})}}$, a generalised Hill function $\beta(v) = \beta_{min} + \frac{\beta_{max}-\beta_{min}}{\left(1+10^{-k(v-v_{50})}\right)^{\frac{1}{b}}}$, a step function with three plateaus and one with four plateaus. We computed the likelihood of each model, searched for the maximum likelihood parameters using the Nelder-Mead method and compared different models based on Akaike Information Criterion (AIC). We tested how transmission varied with other epidemiological factors, including subtype, gender, and circumcision status, by allowing the parameters of the function $\beta(v)$ to vary with different values of these factors (*Figure 1—source data 1*).

## Time to AIDS

The time at which an individual was first diagnosed with AIDS was defined in one of three ways. For the majority of participants, it was defined as the time at which CD4 count is first below 200 cells per mm$^3$, (n = 203 of the 288 participants who declared AIDS) or the time at which three symptoms of AIDS (*Sewankambo et al., 2000*) were first observed (n = 43), whichever came first. If AIDS was not defined according to these criteria, but the individual was known to have died of AIDS, the time to AIDS was taken to be the time to death (n = 42).

Time to AIDS was assumed to follow a gamma distribution whose expected value was a decreasing function of the viral load. For this decreasing function we used a flat function (as a null model), a decreasing Hill function $\hat{t}_{AIDS}(v) = t_{max} \frac{1}{1+10^{-a(v_{50}-v)}}$, a generalised Hill function $\hat{t}_{AIDS}(v) = t_{min} + \frac{t_{max}-t_{min}}{\left(1+10^{-a(v_{50}-v)}\right)^{\frac{1}{b}}}$ and a step function with three plateaus. For the Hill function and the generalised Hill function, we set the maximum time a virus can be carried by its host to $t_{max} = 40$ years. We also allowed these functions to vary by subtype and gender. For a participant, the probability that AIDS occurred between time $t_{no\ AIDS}$ and time $t_{AIDS}$ is:

$$P[t_{no\ AIDS} < t < t_{AIDS}] = \frac{G(k, t_{AIDS}/\theta)}{\Gamma(k)} - \frac{G(k, t_{no\ AIDS}/\theta)}{\Gamma(k)}$$

where $G(k, t_{AIDS}/\theta)/\Gamma(k)$ is the regularized gamma function which is the cumulative distribution function of the gamma distribution; $k$ is the shape parameter and $\theta$ is the scale parameter set to $\hat{t}_{AIDS}/k$ so that the expected value is $\hat{t}_{AIDS}$. When AIDS was not declared in the individual, $t_{no\ AIDS}$ was set to the date of last visit of this individual, and $t_{AIDS}$ was set to infinity. The likelihood function was obtained by multiplying these probabilities across all participants. We computed the likelihood of each model, searched for the maximum likelihood parameters and compared different models based on Akaike Information Criterion (AIC).

## Epidemiological and evolutionary modelling

We developed a Susceptible-Infected compartmental ordinary differential equation (ODE) model, where the viral population is stratified by SPVL. The set-point viral load $v$ of an individual is given by $v=g+e$ where $g$ is the genetic effect, transmitted with mutation from a donor to a recipient, and $e$ is the environmental effect, which includes host and other environmental factors, and is independently drawn in each newly infected individual. The model is akin to classical quantitative genetics models and in particular to a previously described model of virulence evolution (*Lande, 1976*; *Day and Proulx, 2004*). The model neglects the impact on transmission of the higher viral loads in early and late phases of infection, however we relax this assumption in the individual-based model presented below. The number of infected with genetic and environmental effects $(g, e)$ evolves as:

$$\frac{dY(g,e,t)}{dt} = \underbrace{\int_{\gamma=-\infty}^{\infty} \int_{\epsilon=-\infty}^{\infty} \beta(\gamma+\epsilon)X(t)Y(\gamma,\epsilon,t)P(e)Q(\gamma \to g)d\epsilon\, d\gamma}_{transmission} - \underbrace{\mu(g+e)Y(g,e,t)}_{death}$$

and the number of uninfected individuals $X$ changes as:

$$\frac{dX}{dt} = bX - \bar{\beta}XY_{tot}$$

The first term in the equation for the number of infected reflects the increase in the number of infected individuals with viral genetic effect $g$ and environmental effect $e$ due to new transmission events from all possible donors. The second term describes death of infected individuals. In these equations, $\beta(.)$ is the transmission rate as a function of SPVL, $P(e)$ is the distribution of environmental effects in newly infected individuals, $Q(\gamma \to g)$ is the mutation kernel, which is the probability that a donor with virus of genetic effect $\gamma$ gives an infection with a virus of genetic effect $g$, $\mu(.)$ is the AIDS death rate as a function of SPVL (inversely related to the time to AIDS), $b$ is the birth rate, $\bar{\beta}$ is the mean transmission rate in the population, and $Y_{tot}$ is the total number of infected.

The evolution of mean SPVL in the population is determined by the evolution of the mean viral genetic effect $g$, as the mean environmental effect is set at 0 without loss of generality. Under this model, we find that evolution of mean genetic effect (denoted $\bar{g}$) is determined by the Price equation (**Price, 1970**):

$$\frac{d\bar{g}}{dt} = \text{cov}[\beta X - \mu, g] + \alpha\, \bar{\beta}\, X$$

(see SI for derivation). The parameter $\alpha$ is the mean effect of mutations on SPVL in $\log_{10}$ copies/mL. The first term of the equation is the Robertson-Price identity (**Robertson, 1966**; **Price, 1970**), which equates the change in character with the population covariance between a fitness measure, here $\beta X - \mu$, and the genetic value of this character. The dependence on the number of uninfected individuals sets the balance between selection for higher transmission rate and selection for lower mortality. For example, when the number of susceptible individuals is large relative to its long-term equilibrium value $\bar{\mu}/\bar{\beta}$, selection for higher transmission and higher mortality is favored, an effect that can be important in an emerging epidemic (**Bolker et al., 2010**; **Shirreff et al., 2011**; **Berngruber et al., 2013**). The second term describes the effect of biased mutation, proportional to incidence $\bar{\beta}\, X$.

We emphasize that knowledge of the molecular mechanism driving the decline in virulence is not needed to make evolutionary predictions. To derive further analytical insights, we assume that the number of susceptible individuals is approximately at its equilibrium value $\bar{\mu}/\bar{\beta}$. We take advantage of the approximately normal distribution of SPVL in the population to derive an expression for the change in mean SPVL in prevalent cases over time, akin to Lande's classical quantitative genetic equation (**Lande, 1976**).

$$\frac{d\bar{g}}{dt} = V_P\, h^2 \frac{\bar{\mu}^2}{\bar{\beta}} \frac{\partial(\bar{\beta}/\bar{\mu})}{\partial \bar{g}} + \alpha\, \bar{\mu}$$

where $V_P$ is the variance in SPVL and $h^2$ is heritability of SPVL, the fraction of the variance explained by viral genetic factors. The mean SPVL in the population will evolve to the value maximizing mean fitness $\bar{\beta}/\bar{\mu}$, which is 3.4 $\log_{10}$ mL/copies (95% CI [2.6; 4.0], **Figure 2a**), at a pace proportional to heritability (which is assumed to be at equilibrium).

We parameterised the ODE model with our data, and solved it using the Euler method. Specifically, the initial SPVL in incident cases was 4.72 $\log_{10}$ copies/mL. The transmission rate and mortality due to AIDS as a function of SPVL were the inferred functions (**Figure 1**). We tuned the baseline transmission rate and the birth rate to achieve the stable prevalence of 14% observed in the Rakai communities and a total population size of 20 millions adults. Declining prevalence would not change much the evolution of mean SPVL (**Figure 2—figure supplement 5**).

We assumed that the mutation kernel $Q(\gamma \to g)$ was the density of a normal distribution with a non-zero mean $\alpha$, and standard deviation $\sigma_{mut} = 0.15$, evaluated at $g - \gamma$. The density of

environmental effects $P(e)$ was given by the density of a normal distribution with mean 0 and standard deviation 0.76. The variance parameters were chosen to achieve an approximately stable phenotypic variance of SPVL $V_P = 0.91$ and heritability at 36% as inferred in this cohort (*Hollingsworth et al., 2010*), and similar to the value of 30 to 40% established in a number of studies (*Fraser et al., 2014*; *Mitov and Stadler, 2016*).

Because only a small number of studies have linked within-host evolution to SPVL evolution, we explored three scenarios spanning a range of possibilities to parameterise $\alpha$. (i) The dominant process is the increase in the frequency of CTL escape mutations, or other host-specific beneficial mutations imposing a RC cost, resulting in a reduced viral fitness and SPVL in the next typical infected person. We first parameterize $\alpha$ in this scenario using data on the inferred decline in mean SPVL in Botswana (*Payne et al., 2014*). The mean SPVL in a cohort in South Africa was 4.47, compared to 4.19 $\log_{10}$ copies/mL in a cohort in Botswana where the epidemic started about 6 years earlier, giving an inferred decline of (4.19 − 4.47) / 6 = −0.047 $\log_{10}$ copies/mL/year, hypothesized to result from the rise of CTL escape mutations in the viral population. From the Price equation, the decline in mean SPVL is given by $\alpha \, \bar{\mu}$, assuming constant prevalence and neglecting the virulence-transmission trade-off. Solving for $\alpha$ in $\alpha \, \bar{\mu} = -0.047$, with a mean death rate of $\bar{\mu} = 0.1$ per year as in the present cohort, gives a rough estimate of $\alpha = -0.47$ $\log_{10}$ copies/mL under this scenario. (ii) Second, under a similar assumption that the dominant process is the increase in host-specific beneficial mutations imposing a RC cost, we now parameterize $\alpha$ assuming that these mutations impose a RC cost similar to that of random mutations. Indeed some immune escape mutations, for example CTL escape mutations arising in the pol, env or nef gene, appear neutral (*Matthews et al., 2008*; *Troyer et al., 2009*). In this scenario, the coefficient of variation of the distribution of SPVL effects within the host would be the same as that of the distribution of fitness effects of random mutations. This coefficient of variation was estimated at −1.609 in a previous study (*Bonhoeffer et al., 2004*), giving $\alpha = -\sigma_{mut}/1.609 = -0.093$ $\log_{10}$ copies/mL. (iii) The dominant process is the increase in frequency of mutations causing a within-host increase in RC, resulting in higher viral fitness in the next host. To our knowledge increase in RC over the course of infection has been evidenced only in one study (*Kouyos et al., 2011*). This study predicted an increase in RC over the course of infection of + 0.02 per year. The relationship between RC and SPVL inferred in that study (SPVL = 4.297 + 0.572 * RC, *Figure 1A* in [*Kouyos et al., 2011*]), together with the fact that the mean time to transmission is 5 years (as inferred from simulation of our IBM), leads to $\alpha = 0.02 \ \times 5 \ \times \ 0.572 = + 0.057$ $\log_{10}$ copies/mL in this scenario.

Predictions of the ODE model were robust to the addition of a number of more realistic features of the HIV epidemic, as shown by an individual-based stochastic model of HIV evolution (IBM) with a higher level of complexity, described in details previously (*Herbeck et al., 2014*; *Herbeck et al., 2016*). The IBM relaxed several assumptions of the ODE. In contrast to the ODE that described only the asymptomatic phase of infection characterized by a stable SPVL value, the IBM explicitly modelled the dynamics of viral load within individuals. This included the acute phase of infection and the AIDS phase, which are both characterized by a higher viral load. The viral load in the acute and AIDS phases, and the duration of acute phase did not vary across individuals. In the ODE, transmission was modelled using the law of mass action; in the IBM a changing network of sexual contacts was modelled (although sexes were not explicitly modelled). The number of partnerships in which each individual was engaged was variable, and there was heterogeneity in partnership duration (between 3 and 60 months). Furthermore, the behavioural dynamics were designed to reflect a core group of transmitters; individuals in the core group (10% of the overall population) had shorter partnership durations and increased coital frequency. The rate of overall partnership formation and the distribution of coital frequencies were both calibrated to result in an equilibrium prevalence of 14%, corresponding to the average prevalence in the 1995–2015 period, as for the main model.

## Temporal trends in SPVL

We inferred temporal trends in SPVL in incident cases using a multivariate linear model where we explained variation in SPVL as a function of the laboratory in which SPVL was measured, the assay used, whether VL was measured at a RCCS visit (individuals with unclear infection status), gender, circumcision status, age, date at seroconversion, and subtype (*Figure 2*). Significance was assessed using type II analysis of variance, and confidence intervals were computed assuming asymptotic normality of the coefficients. Viral loads were measured in three different laboratories and using two

types of PCR assays. This heterogeneity of laboratory approaches could potentially confound other trends; however our multivariate regression controlled for these effects, and revealed that they had small and non-significant effect sizes (*Figure 2—source data 1*), such that they did not generate any systematic variability in SPVL. SPVL decreased at a pace of $-0.033$ $\log_{10}$ unit per year (CI $[-0.057;$ $-0.009]$, p=0.007, n = 603), resulting in a 0.66 $\log_{10}$ unit change over the 1995–2015 period. The estimated rate was $-0.022$ (CI $[-0.041;$ $-0.002]$, p=0.027, n = 603) after non-significant predictors were removed. The linear temporal trend in mean SPVL was more supported than a model where time was fitted as five discrete categories ($\Delta$AIC = 7.2). An important potential confounder of the reported trends in SPVL would have been the use of unreported antiretroviral therapy (ART) becoming more frequent at later time points. To exclude this possibility, we focused on the subset of SPVL values with viral loads measured before 2004, prior to ART availability in the region. Consistent with previous studies (*Farzadegan et al., 1998*; *Gandhi et al., 2002*), males had a higher SPVL than females (+0.259 $\log_{10}$ viral copies/mL, CI [0.14; 0.38], p=4.2 $10^{-5}$, n = 603) subtype D conferred higher SPVL than other subtypes (+0.211 relative to subtype A, CI [0.038; 0.38], p=0.017, n = 603), and older age conferred slightly higher SPVL (+ 0.009 per year, CI [0.0008; 0.016], p=0.030, n = 603). The decreasing trend in SPVL as well as the effects of gender, and subtype D, were all robust, as they had similar magnitude in several subsets of data (*Figure 2—figure supplement 2*).

We also inferred temporal trends in mean SPVL in prevalent cases by calculating each year the mean SPVL for cases who are infected, alive, and not lost to follow-up. In this analysis we found a decline in mean SPVL at a rate of $-0.020$ $\log_{10}$ copies/mL/year (*Figure 2—figure supplement 6*). This decline was highly significant (p=5.06e−08, N = 18) but the p-value calculation did not account for non-independence across years (the same prevalent cases may be included in multiple years).

## Review of coinfections as potential confounders of the SPVL trend

Coinfections such as tuberculosis, malaria, the herpes simplex virus 2, gonorrhea, or syphilis, might increase viral load in HIV infected individuals (*Modjarrad and Vermund, 2010*). A reduction in prevalence $\delta p$ of a disease with an effect $\delta v$ on SPVL would cause a $\delta p \, \delta v$ decrease in mean SPVL in the population. We systematically reviewed these diseases and show that potential reduction in prevalence of these diseases is unlikely to cause the observed 0.4 $\log_{10}$ copies/mL decline in mean SPVL.

Tuberculosis results in a $\delta v = 0.5$ $\log_{10}$ copies/mL increase in viral load (*Modjarrad and Vermund, 2010*), prevalence has decreased two-fold since 1995, and was 2.7% in 2014 among HIV infected persons screened for TB (*WHO, 2015*). This would result in a change in SPVL $\delta p \, \delta v = -0.027 * 0.5 = -0.013$ $\log_{10}$ copies/mL. Malaria incidence is high in Uganda (50.8 episodes per 100 person years in Uganda in 2001, [*Mermin et al., 2006*]), but malaria infection only causes a transient increase in SPVL of $\delta v = 0.25$ $\log_{10}$ copies/mL during ~ 40 days (*Kublin et al., 2005*). The overall effect of a hypothetical two-fold reduction in malaria incidence from 1995 to 2012 (from 60 to 30 per 100 person years) would be $\delta p \, \delta v = -0.3 * 40/465 * 0.25 = -0.006$ $\log_{10}$ viral copies per mL. Herpes simplex virus 2 (HSV-2) prevalence was roughly stable, from 70% in 1994–1998 (*Serwadda et al., 2003*) to 88% in 2007–2008 (*Reynolds et al., 2012*) in HIV infected individuals in the Rakai district, and the prevalence of genital ulcer disease in the general populations, mostly caused by HSV-2 (*Brankin et al., 2009*) was stable over this period (data not shown). The prevalence of gonorrhea and syphilis was 8.6% and 3.3% respectively in 1994–1998 (*Ahmed et al., 2001*); therefore, given these diseases confer $\delta v = 0.04$ and $\delta v = 0.1$ $\log_{10}$ copies/mL increase in HIV viral load (*Modjarrad and Vermund, 2010*), an hypothetical two-fold reduction of prevalence from 1995 to 2012 would have caused a $-0.043 * 0.04 = -0.0018$ $\log_{10}$ viral copies per mL and $-0.017 * 0.1 = -0.0017$ $\log_{10}$ viral copies per mL. Last, coinfection by helminths is rare in most of the Rakai communities (*Wawer et al., 1999*), although schistosomiasis is endemic in some fishing communities living near lake Victoria, with prevalence of up to 50% in 1998–2002 (*Kabatereine et al., 2004*). However, there is no evidence for an effect of helminth infection on HIV viral load (*Brown et al., 2004*; *Modjarrad et al., 2005*; *Modjarrad and Vermund, 2010*).

## Subtype-specific predictions

We extended the ODE model to account for subtype-specific dynamics, in particular the dynamics of subtype A, subtype D, and AD recombinants (called 'R'). The functions describing transmission as a function of SPVL were the inferred subtype-specific step functions (*Figure 3a*). The function

describing time to AIDS as a function of SPVL was the step function inferred on the whole cohort, as there was little difference between subtypes (*Figure 1c*). Starting conditions were parameterised based on the data, as follows. Mean SPVL in incident cases in 1995 were $\bar{v}_{A,0} = 4.58$, $\bar{v}_{D,0} = 4.79$, $\bar{v}_{R,0} = 4.66$ $\log_{10}$ copies per mL of blood. The frequencies of the three types in 1995 were $p_A$=0.17, $p_D$=0.7, $p_R$=0.13. The mutation kernel $Q(\gamma \rightarrow g)$ was, for all three types, the density of a normal distribution with a non-zero mean $\alpha = -0.093$ (scenario 2), and standard deviation $\sigma_{mut} = 0.15$, evaluated at $g - \gamma$. The density of environmental effects $P(e)$ was the density of a normal distribution with mean 0 and standard deviation 0.67. These parameters were chosen to achieve an approximately stable phenotypic variance of SPVL $V_P = 0.7$ (the phenotypic variance in SPVL within subtype) and heritability at 36%.

We assumed super-infection occurred on a fast timescale and immediately resulted in one strain replacing the other. Super-infection with A and D, A and R, or D and R strains resulted in a recombinant subtype ('R') with probability $r$. We chose $r$=1 as it best reproduced the rise in frequency of recombinants (*Figure 3d*).

## Contributions of within-subtype and between-subtype evolution to SPVL trends

We decomposed the trend in mean SPVL into the sum of two components, one due to changes in subtype frequency, one due to within-subtype changes in SPVL. The change in mean SPVL between time 0 and $t$ reads:

$$\Delta\bar{v} = \sum_{i \in \{A, D, R\}} p_{i,t}\bar{v}_{i,t} - \sum_{i \in \{A, D, R\}} p_{i,0}\bar{v}_{i,0}$$

With linear trends in subtype frequencies, $p_{i,t} = p_{i,0} + \delta p_i\, t$, and in mean SPVL within subtypes, $\bar{v}_{i,t} = \bar{v}_{i,0} + \delta\bar{v}_i\, t$. Replacing yields:

$$\Delta\bar{v} = \sum_{i \in \{A, D, R\}} \left(p_{i,0} + \delta p_i\, t\right)\left(\bar{v}_{i,0} + \delta\bar{v}_i\, t\right) - \sum_{i \in \{A, D, R\}} p_{i,0}\bar{v}_{i,0}$$

Because the changes are slow (i.e. $\delta p_i$ and $\delta\bar{v}_i$ are small), we can neglect the term in $\delta p_i \delta\bar{v}_i$ and approximate the change as:

$$\Delta\bar{v} = \left[\sum_{i \in \{A, D, R\}} p_{i,0}\delta\bar{v}_i + \sum_{i \in \{A, D, R\}} \delta p_i\, \bar{v}_{i,0}\right] t$$

The first term reflects the changes in mean SPVL due to changes in mean SPVL within subtype. The second term reflects the changes in mean SPVL due to changing subtype frequencies. We have $\bar{v}_{A,0} = 4.58$, $\bar{v}_{D,0} = 4.79$, $\bar{v}_{R,0} = 4.66$ $\log_{10}$ copies/mL, and $\delta p_A = 0.009$, $\delta p_D = -0.016$, $\delta p_R = 0.007$, inferred from a generalized linear model with multinomial response describing subtype frequency as a function of calendar time. Thus the change in mean SPVL due to the rise in subtype A and R is $-0.003$ $\log_{10}$ copies/mL per year. Assuming the same rate of SPVL evolution in all subtypes, $\delta\bar{v}_A = \delta\bar{v}_D = \delta\bar{v}_R = -0.022$ $\log_{10}$ copies/mL per year (a rate inferred from the linear model, adjusted for subtype and other covariates), the change in mean SPVL due to within-host evolution is also $-0.022$ $\log_{10}$ copies/mL per year. Thus the total mean SPVL change is $= -0.025$ $\log_{10}$ copies/mL per year and most of this change is due to within-subtype evolution.

## Acknowledgements

We thank Troy Day, Florence Débarre, Sylvain Gandon, Prabhat Jha, Richard Neher and an anonymous reviewer for useful comments. FB is supported by a Marie Skłodowska-Curie Individual Fellowship (grant number 657768). JTH is supported by grants from the U.S. National Institutes of Health (R01AI108490 to JTH, and P30AI027757 to the University of Washington Center for AIDS Research). CF is supported by the European Research Council (Advanced Grant PBDR-339251). This work is supported by the National Institutes of Health (NIH), National Institute of Allergy and Infectious Diseases (NIAID) (grant R01 AI 29314, R01 AI34826); the NIH, NIAID, Division of AIDS, and in part the NIH, NIAID, Division of Intramural Research (grant U01 AI11171-01-02); the National Institute of

Child Health and Development, Johns Hopkins Population Center (grant 5P30 HD 06268); the Fogarty Foundation (grant 5D43TW00010); John Snow Inc, Pfizer Inc (grant 5024-30); the Rockefeller Foundation; the World Bank STI Project, Uganda; a cooperative agreement (W81XWH-07-2-0067) between the Henry M. Jackson Foundation for the Advancement of Military Medicine, Inc., and the U.S. Department of Defense (DOD). The views expressed in this article are those of the author and do not necessarily reflect the official policy or position of the Department of Defense, nor the US Government.

## Additional information

### Funding

| Funder | Grant reference number | Author |
| --- | --- | --- |
| European Commission | Intra European Fellowship 657768 | François Blanquart |
| National Institute of Allergy and Infectious Diseases | R01 AI29314 | Mary Kate Grabowski<br>Fred Nalugoda<br>David Serwadda<br>Michael A Eller<br>Merlin L Robb<br>Ronald Gray<br>Godfrey Kigozi<br>Oliver Laeyendecker<br>Gertrude Nakigozi<br>Thomas C Quinn<br>Steven J Reynolds<br>Maria J Wawer |
| National Institute of Child Health and Development | 5P30 HD 06268 | Mary Kate Grabowski<br>Fred Nalugoda<br>David Serwadda<br>Michael A Eller<br>Merlin L Robb<br>Ronald Gray<br>Godfrey Kigozi<br>Oliver Laeyendecker<br>Gertrude Nakigozi<br>Thomas C Quinn<br>Steven J Reynolds<br>Maria J Wawer |
| John E. Fogarty Foundation for Persons with Intellectual and Developmental Disabilities | 5D43TW00010 | Mary Kate Grabowski<br>Fred Nalugoda<br>David Serwadda<br>Michael A Eller<br>Merlin L Robb<br>Ronald Gray<br>Godfrey Kigozi<br>Oliver Laeyendecker<br>Gertrude Nakigozi<br>Thomas C Quinn<br>Steven J Reynolds<br>Maria J Wawer |
| John Snow Inc. | 5024-30 | Mary Kate Grabowski<br>Fred Nalugoda<br>David Serwadda<br>Michael A Eller<br>Merlin L Robb<br>Ronald Gray<br>Godfrey Kigozi<br>Oliver Laeyendecker<br>Gertrude Nakigozi<br>Thomas C Quinn<br>Steven J Reynolds<br>Maria J Wawer |
| Pfizer | 5024-30 | Mary Kate Grabowski<br>Fred Nalugoda<br>David Serwadda |

| Funder | Grant reference number | Author |
|---|---|---|
| | | Michael A Eller |
| | | Merlin L Robb |
| | | Ronald Gray |
| | | Godfrey Kigozi |
| | | Oliver Laeyendecker |
| | | Gertrude Nakigozi |
| | | Thomas C Quinn |
| | | Steven J Reynolds |
| | | Maria J Wawer |
| Rockefeller Foundation | | Mary Kate Grabowski |
| | | Fred Nalugoda |
| | | David Serwadda |
| | | Michael A Eller |
| | | Merlin L Robb |
| | | Ronald Gray |
| | | Godfrey Kigozi |
| | | Oliver Laeyendecker |
| | | Gertrude Nakigozi |
| | | Thomas C Quinn |
| | | Steven J Reynolds |
| | | Maria J Wawer |
| World Bank Group | | Mary Kate Grabowski |
| | | Fred Nalugoda |
| | | David Serwadda |
| | | Michael A Eller |
| | | Merlin L Robb |
| | | Ronald Gray |
| | | Godfrey Kigozi |
| | | Oliver Laeyendecker |
| | | Gertrude Nakigozi |
| | | Thomas C Quinn |
| | | Steven J Reynolds |
| | | Maria J Wawer |
| National Institute of Allergy and Infectious Diseases | R01 AI34826 | Mary Kate Grabowski |
| | | Fred Nalugoda |
| | | David Serwadda |
| | | Michael A Eller |
| | | Merlin L Robb |
| | | Ronald Gray |
| | | Godfrey Kigozi |
| | | Oliver Laeyendecker |
| | | Gertrude Nakigozi |
| | | Thomas C Quinn |
| | | Steven J Reynolds |
| | | Maria J Wawer |
| National Institute of Allergy and Infectious Diseases | U01 AI11171-01-02 | Mary Kate Grabowski |
| | | Fred Nalugoda |
| | | David Serwadda |
| | | Michael A Eller |
| | | Merlin L Robb |
| | | Ronald Gray |
| | | Godfrey Kigozi |
| | | Oliver Laeyendecker |
| | | Gertrude Nakigozi |
| | | Thomas C Quinn |
| | | Steven J Reynolds |
| | | Maria J Wawer |
| National Institutes of Health | P30AI027757 | Joshua Herbeck |
| National Institutes of Health | R01AI108490 | Joshua Herbeck |
| U.S. Department of Defense | W81XWH-07-2-0067 | Michael A Eller |
| | | Merlin L Robb |
| Henry M. Jackson Foundation | W81XWH-07-2-0067 | Michael A Eller |
| | | Merlin L Robb |
| European Research Council | PBDR-339251 | Christophe Fraser |

The funders had no role in study design, data collection and interpretation, or the decision to submit the work for publication.

## Author contributions
FB, Designed the study, analysed the data, did the modelling and simulations, wrote the draft; MKG, Acquisition of data, Analysis and interpretation of data, Drafting or revising the article; JH, KAL, Analysis and interpretation of data, Drafting or revising the article; FN, DS, GK, GN, Acquisition of data; MAE, MLR, RG, OL, TCQ, SJR, MJW, Acquisition of data, Drafting or revising the article; CF, Conception and design, Analysis and interpretation of data, Drafting or revising the article

## Author ORCIDs
François Blanquart, http://orcid.org/0000-0003-0591-2466
Joshua Herbeck, http://orcid.org/0000-0003-4577-7406

## Ethics
Human subjects: Informed consent was obtained from all the participants in the Rakai Community Cohort Study. The Scientific and Ethics Committee of the Uganda Virus Research Institute (UVRI) of the Ministry of Health provides the Institutional Review Board approval and monitoring of all Rakai research.

# Additional files

## Supplementary files
• Supplementary file 1. Individual viral load trajectories within patients for 603 incident cases with a SPVL value (when undetectable viral load were removed from the SPVL calculation). Points are viral load values, shown as solid bullets when used for the SPVL calculation, and open circles otherwise. The vertical red line is the mid-point between last negative test and first positive test. The vertical light green line is the date ART started. The vertical dark green line is the date of first self-reported ART. The horizontal black line is the SPVL value. This data relates to *Figure 2*.

• Reporting standard 1. Summary table of statistical tests.

## Major datasets
The following dataset was generated:

| Author(s) | Year | Dataset title | Dataset URL | Database, license, and accessibility information |
|---|---|---|---|---|
| Blanquart F, Grabowski MK, Herbeck J, Nalugoda F, Serwadda D, Eller MA, Robb ML, Gray R, Kigozi G, Laeyendecker O, Lythgoe K, Nakigozi G, Quinn TC, Reynolds SJ, Wawer MJ, Fraser C | 2016 | Data from: A transmission-virulence evolutionary trade-off explains attenuation of HIV-1 in Uganda | http://dx.doi.org/10.5061/dryad.7kr85 | Available at Dryad Digital Repository under a CC0 Public Domain Dedication |

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

## Appendix 1

# Details of the data analysis and the model

The Rakai Community Cohort Study (RCCS) is a population-based cohort of HIV incidence and sexual behaviours set in the Rakai district, Uganda conducted on an approximately annual basis since 1994. The RCCS survey obtains detailed information on demographics, sexual behaviours and health status and specimens for HIV testing and other research purposes. Prior to 2013, the Rakai Health Sciences Program (RHSP), which administers the RCCS, primarily managed HIV care and treatment in the Rakai District and so these clinical data could be linked back to RCCS for research purposes. We used two subsets of this data. First, to investigate the temporal trends in SPVL and the relationship between SPVL and time to AIDS, we used the 'incident cases', defined as participants with a first positive result within two RCCS study visits of their last negative result, such that the date of infection is known relatively precisely. Second, to investigate the relationship between SPVL and transmission rate, we identified the subset of participants engaged in serodiscordant partnership. The 'serodiscordant couples' are defined as those couples where one partner is positive, while the other is initially negative and may become infected over the course of follow-up.

## Data cleaning and set-point viral load calculations

### Incident cases

To investigate the time trends in SPVL and the relationship between SPVL and time to AIDS, we used the incident cases, defined as participants with a first positive result within two RCCS study visits of their last negative result. The median time elapsed between last negative and first positive result was 1.25 year (minimum 0.56, maximum 3.59 years).

### SPVL calculation

The date of seroconversion for each participant was defined as the mid-point between the last HIV-negative date and the first HIV-positive date. The SPVL was defined as the mean $\log_{10}$-viral load at all visits occurring more than 6 months after seroconversion, and before the beginning of antiretroviral therapy (ART).

ART was prescribed in the Rakai district from 2004 onwards. We had clinical records on the date of ART start when it was prescribed in an RHSP clinic prior to 2013. However, participants may also have received ART from clinics outside of RHSP. Moreover, by 2013, ART distribution at the majority of RHSP clinics was transferred to the Ugandan Ministry of Health. To determine whether participants were prescribed ART outside of RHSP clinics, we relied on self-reported use of ART. For these cases, we defined the date of self-reported ART start as the mid-point between the last date no ART was reported and the first date where ART was reported. We further removed all viral load measures at a date later than this self-reported ART start date. 2135 viral load measures out of the 5180 were taken 6 months after the date of seroconversion and before ART.

A number of viral loads measures were below detection limit of the assay (<400 copies/mL), in which case the viral load is reported as 'undetectable' and we did not know its precise value. Such low viral loads could mean these participants are 'elite controllers'; but they could also be due to measurement error and/or degradation of the RNA sample, participants having been prescribed ART in a clinic other than the RHSP clinics without reporting it, or participants falsely believed to be infected. We assumed that any single viral load measure below 400 copies/mL in a participant with more than one measure over 400 copies/mL was due to either error or sample degradation, and we removed these measurements from our analysis (44 measures out of 2135). We verified that the HIV infection status of participants by reassessing all available serological HIV tests results, including rapid assays, ELISAs, and Western Blot assays. We removed from the analysis any

participants whose infection status was unclear, i.e. those participants who only have 'undetectable' viral load and who either have at least one 'indeterminate' or 'negative' Western Blot and two or less ELISA tests, or who have no Western Blot and at least one negative ELISA test (7 measures out of 2135). All remaining 'undetectable' viral loads (n = 2071) were set to 200 copies/mL. We systematically verified the robustness of our analysis to the inclusion of the 'undetectable' viral loads.

Because of the uncertainty on the timing of infection, it is possible that a participant is still in acute infection more than 6 months after the presumed date of infection (the mid-point). To avoid this possibility, we also eliminated first measures where the viral load was ten times greater than all subsequent measures (13 measures out of 2135).

Following these steps of data cleaning, we obtained SPVL for 647 individuals, each SPVL measure representing 1 to 16 visits (median = 2) and a total of 2071 viral load measures. We also computed a SPVL value where all 'undetectable' viral loads are discarded, in which case we obtain SPVL for 603 individuals, each SPVL measure representing 1 to 16 visits (median = 4).

## Subtyping

Subtype of each participant was determined using one or several of four different methods, based on (i) sequence fragments of gp41 and p24, (ii) Roche 454 sequencing of gp41 and p24, (iii) multi-region hybridization assays on gag, pol, vpu, env, and gp41 and (iv) full genome sequences (*Kiwanuka et al., 2008*; *Conroy et al., 2010*). All subtype information was compiled for each participant, by genomic region (gag, pol, vpu, and env). In 467 cases out of the 576 participants with any subtype information, all subtype information agreed on a single subtype, which was then assigned to the participant. If the data indicated infection with multiple HIV subtypes (i.e. one or more HIV subtypes detected in the same gene region), we assigned the subtype 'multiple' (M) to the individuals. Lastly, if HIV-1 subtype differed across but not within genes, a recombinant subtype was assigned to the participant. This algorithm resulted in 12 subtype categories, of which the most represented in the population were subtype D (n=342) subtype A (n=118), recombinant DA (n=41), recombinant AD (n=28), multiple infections M (n=17), subtype C (n=8) and various types of recombinants (n=22). A linear regression of SPVL against subtype revealed that a more parsimonious model with only five simplified subtype categories, A, C, D, M, R (all recombinants) was a better fit to the data than a model with all 12 subtype categories ($\triangle$AIC = 5.5).

## Viral load laboratories and assays

Viral RNA was quantified in one of three laboratories: the Makerere University Walter Reed Project Laboratory (Kampala, Uganda) (WR), the International HIV and STD laboratory at Johns Hopkins University (Baltimore, MD, USA) (JH), or at the Rakai Health Sciences Program central laboratory (Kalisizo, Uganda) (RHSP). At WR and JH, all assays were conducting using the Roche Amplicor v1.5 assay. RHSP used the Roche Amplicor v1.5 from May 2005 to Nov 2010 and the Abbott m2000 from October 2010 up to date. In order to test for potential effect of assay on viral load values, we assume the date at which each sample was assayed is approximately the date of the sample. Last, when the infection status of the participants was unclear from the serology results, viral loads were measured in the RHSP laboratory on samples taken at an RCCS visit; we included these particular samples as a variable in the regression to control for potentially lower SPVL for these participants.

## Time to AIDS

The time to AIDS was the time at which CD4 count is first below 200 cells per mm$^3$, three symptoms of AIDS were first observed, or of AIDS death. The time to AIDS is a reasonable approximation of the time during which the virus will be transmitted. Transmission could be also interrupted because of host death not related to HIV, but AIDS was the most common cause of mortality in Uganda among 13 - 44 years (*Mulder et al., 1994*; *Sewankambo et al., 2000*). Moreover, there is little opportunity for transmission after AIDS is declared, as host

death occurs shortly after the onset of AIDS (median 1.46 years) and it has been estimated that very little transmission occurs in the 10 months prior to death (*Hollingsworth et al., 2008*).

### Serodiscordant couples

To investigate the relationship between SPVL and transmission rate, we focused on retrospectively identified serodiscordant couples, a subset of data largely distinct from the incident cases. In these couples, one partner is positive, while the other is initially negative and may become infected over the course of follow-up. The median duration of couple follow-up was 2.6 years. As before, we set 'undetectable' viral loads to 200 copies/mL, and the SPVL was defined as the mean $\log_{10}$-viral load at all visits occurring more than 6 after seroconversion, and before the beginning of antiretroviral therapy (ART). We also calculated a SPVL with undetectable values removed to check that this small number of imprecise measures at low viral loads did not affect inference of transmission rates.

Subtypes were computed as before, except we simplified further the categories to keep only subtype A, subtype D, and 'Other/unknown' subtypes (including other subtypes, multiple infections, recombinants, and unknown).

Our analysis data set included 817 couples with SPVL values from the index HIV-infected partner. Each SPVL represented 1 to 15 measures (median = 2). The SPVL data from these couples are summarized in *Table 2*.

## Inference of temporal trends in SPVL

The variability in the number of measures underlying SPVL poses several problems for the analysis of temporal trends in SPVL. There was no effect of the number of measures on SPVL (ANOVA with number of visits as factor, *p = 0.61*, n = 603), but the *variance* of SPVL across individuals was much higher among individuals measured once than across those measured multiple times. This is expected as multiple measures reduce the effects of intra-individual fluctuations in VL and measurement error on SPVL. This means the error around each SPVL value was not the same across individuals, the true SPVL being approached much more closely in individuals with many measures than in individuals with a single measure. When analysing the determinants of SPVL, this will cause heteroscedasticity, violating an assumption of linear models. To overcome this problem, it has been proposed to fit the viral load trajectories within individual using a fractional polynomial within a mixed model, then to regress several descriptors of this fractional polynomial over the predictors of interest (*Pantazis et al., 2014*). This method cannot easily include multiple predictors, and is clearly not applicable in our case where 242 out of 526 participants have one viral load measurement only. To verify that our results were not affected by this problem, we also conducted an analysis on the subset of SPVL values which include 2 measurements or more and where the standard deviation across these is less than 1 log-viral load unit across measures ('strict SPVL').

### Linear model

We used a linear regression to explain the variation in SPVL across participants as a function of gender, age at seroconversion, subtype, and date of seroconversion, using the data on the incidence cases for whom the date of seroconversion was known precisely. We also tested the effect of being circumcised in males. We corrected for potential confounding due to the use of several viral load assays and measurements in several laboratories by including these factors in the regression.

We first analysed the determinants of SPVL being 'undetectable' (that is, those participants where all viral load measures are undetectable), using a logistic regression over epidemiological covariates. Next we analysed the determinants of SPVL on the subset of data that excludes undetectable SPVL, because these SPVL values caused the distribution of SPVL to be non-normal, violating an assumption of linear models (the 'undetectable' were set at $\log_{10}(200)$). To verify the robustness of our predictions, we ran the analysis on several

subsets of data: (i) SPVL calculated from at least two viral load measures, with a standard deviation of less than 1 log-viral load unit across measures, and including undetectable viral loads ('strict SPVL') – a subset for which the heteroscedasticity problem will be reduced -, (ii) data partitioned in two subsets corresponding to the assays Abbott m2000 and Roche 1.5 (iii) data partitioned in three subsets corresponding to the laboratories (John Hopkins, RHSP, Walter Reed), (iv) data partitioned in male and female.

The distribution of SPVL was visually very close to a normal distribution. However the Shapiro - Wilk test of normality rejected the normal distribution (n = 603, p = 0.0018), in particular because of an excess of low SPVL.

The probability that the SPVL is 'undetectable' was mainly determined by the assay, Abbott m2000 giving an 'undetectable' SPVL with higher probability than Roche 1.5 (*Appendix 1—table 1*, n = 647). There was no effect of subtype, except that individuals whose subtype is unknown also tended to have 'undetectable' SPVL with higher probability, because a low viral load made subtyping harder.

**Appendix 1—table 1.** Summary of adjusted effects and p-values obtained by type II analysis of deviance for the logistic regression of 'detectable versus undetectable SPVL' over epidemiological covariates. n = 647. 'p < 0.1, *p<0.05, ** p<0.01, ***p< 0.001.

| Factor | Effect size | p-value |
|---|---|---|
| Intercept | 0.217 | 0.002 ** |
| John Hopkins | Reference | - |
| RHSP | −0.029 | 0.354 |
| Walter Reed | −0.041 | 0.117 |
| Abbott | Reference | - |
| Roche 1.5 | −0.158 | 0.001 ** |
| Not RCCS visit | Reference | - |
| RCCS visit | 0.01 | 0.747 |
| Female | Reference | - |
| Male | −0.021 | 0.261 |
| Age | 0.001 | 0.378 |
| Date seroconversion | −0.005 | 0.149 |
| Subtype A | Reference | - |
| Subtype C | −0.012 | 0.899 |
| Subtype D | −0.011 | 0.693 |
| Recombinant | 0.002 | 0.948 |
| Dual infections | 0.051 | 0.434 |

The laboratory where the viral load was assayed and the assay used had a small effect on SPVL. The inclusion of VL values measured at a RCCS visit ('RCCS visit' in *Figure 2—source data 1*) led to significantly lower SPVL (-0.3 $\log_{10}$ copies/mL, CI [-0.51; -0.09], p = 0.006, n = 603), which was expected as viral loads were measured at a RCCS visit only when the infection status of the participants was unclear from the serology results. Variation in SPVL was mainly predicted by gender and date of seroconversion (*Figure 2—figure supplement 2*).

## Fitting the transmission rate as a function of SPVL

Using data on transmission and SPVL in 817 serodiscordant couples, we estimated the transmission hazard as a function of SPVL (Methods). The transmission hazard is the expected number of transmission events per unit time in a serodiscordant couple. This revealed that transmission increased significantly with SPVL, and the best functional form

to describe this relationship was a step function with three plateaus. The best relationship was robust to the presence or absence of 'undetectable' SPVL in the dataset (*Figure 1—figure supplement 2*). We also tested whether transmission varied by subtype, gender, and whether the male was circumcised or not.

### Fitting the time to AIDS as a function of SPVL

The time to AIDS was assumed to follow a gamma distribution whose expected value was a decreasing function of the viral load (Materials and methods). We derived the likelihood function, found maximum likelihood parameters using the Nelder-Mead method, and compared the different models based on AIC. The step function was the one that described best the relationship between time to AIDS and SPVL (*Figure 1*). We tested how several covariates affected time to AIDS including subtype, and gender, but none of them significantly improved the fit. The best relationship was robust to the presence or absence of 'undetectable' SPVL in the dataset (*Figure 1—figure supplement 2*).

### A quantitative genetics model for the evolution of set-point viral load

#### Analysis of a quantitative genetics model

We consider the SPVL as a viral trait that evolves to maximize transmission. We developed a compartmental model describing the dynamics of susceptible and infected individuals in the population. Each infected individual has SPVL $v = g + e$, where $g$ is the breeding value of SPVL and $e$ is the environmental effect. The breeding value $g$ is transmitted almost perfectly from the one infection to the next, except it can be modified by mutations. The environmental effect, in contrast, is independently drawn at each new infection. The model is similar to a previous model on the evolution of virulence (*Day and Proulx, 2004*), except we consider the effect of the environment on SPVL and the effect of mutation on $g$ is modelled more generally (a diffusion model was used previously [*Day and Proulx, 2004*]). The evolution of the number of infected with genetic and environmental values $(g, e)$ is given by:

$$\frac{dY(g,e,t)}{dt} = \int_{\gamma=-\infty}^{\infty} \int_{\epsilon=-\infty}^{\infty} \beta(\gamma+\epsilon)X(t)Y(\gamma,\epsilon,t)P(e)Q(\gamma \to g)d\epsilon\, d\gamma \; - \mu(g+e)Y(g,e,t)$$

The first term represents infected individuals with viral genotypic value and environmental value $(\gamma, \epsilon)$ ('donors') infecting susceptible individuals $X(t)$ ('recipients'). The newly infected individual will carry a virus $(g, e)$ with probability $P(e)\, Q(\gamma \to g)$ where $P(e)$ is the probability that the new environmental effect is $e$ and is given by the density of a normal distribution with mean 0 and variance $\sigma_e^2$, and $Q(\gamma \to g)$ is the probability that the recipient has genetic value $g$ given that the donor has genetic value $\gamma$, and is given by the density of a normal distribution with mean $\alpha$ and variance $\sigma_{mut}^2$, evaluated at $g - \gamma$. The parameter $\alpha$ quantifies the mutational bias, and is equal to 0 if on average mutations do not affect SPVL. The second term represent death of individuals due to infection.

Heritability, defined as the regression coefficient of the recipient's viral load regressed against the donor's viral load, is equal to $\frac{V[G]}{V[G]+V[E]}$ where $V[G]$ is the population variance in $g$, $V[E]$ is the population variance in $e$. While it is difficult to predict the exact value of heritability as a function of parameters of the model, the genetic variance $V[G]$ will increase with mutational variance $\sigma_{mut}^2$ and the environmental variance $V[E]$ will increase with the variance of the environmental effect $\sigma_e^2$.

The model neglects the contribution of acute infection or AIDS in transmission. This is justified, as the asymptomatic phase was the largest contributor to transmission in the Rakai cohort (*Hollingsworth et al., 2008*), and recent work suggests the relative infectivity in the early phase of infection has previously been overestimated, such that the contribution of early phase to total transmission could be about 7 times smaller than that of the asymptomatic phase (*Bellan et al., 2015*).

$Y_{tot}(t)$ is the total number of infected given by:

$$Y_{tot}(t) = \int\limits_{g=-\infty}^{\infty} \int\limits_{e=-\infty}^{\infty} Y(g,e,t) dg\, de$$

the total number of infected evolves as (dependence on time is dropped for clarity):

$$\frac{dY_{tot}}{dt} = \int\limits_{g=-\infty}^{\infty} \int\limits_{e=-\infty}^{\infty} \int\limits_{\gamma=-\infty}^{\infty} \int\limits_{\epsilon=-\infty}^{\infty} \beta(\gamma+\epsilon) X\, Y(\gamma,\epsilon) P(e) Q(\gamma \to g) d\epsilon\, d\gamma dg\, de$$
$$- \int\limits_{g=-\infty}^{\infty} \int\limits_{e=-\infty}^{\infty} \mu(g+e) Y(g,e) dg\, de$$

Because $\int\limits_{g=-\infty}^{\infty} Q(\gamma \to g) dg = 1$ and $\int\limits_{e=-\infty}^{\infty} P(e) de = 1$, this simplifies to:

$$\frac{dY_{tot}}{dt} = \bar{\beta}\, X\, Y_{tot} - \bar{\mu}\, Y_{tot}$$

with $\bar{\mu}(t) = \int\limits_{g=-\infty}^{\infty} \int\limits_{e=-\infty}^{\infty} \mu(g+e)\, \phi(g,e) dg\, de$

and $\bar{\beta}(t) = \int\limits_{g=-\infty}^{\infty} \int\limits_{e=-\infty}^{\infty} \beta(g+e)\, \phi(g,e) dg\, de$

where $\phi(g,e) = Y(g,e)/\, Y_{tot}$ is the frequency of each infected type in the population. The number of susceptible evolves as:

$$\frac{dX}{dt} = b\, X - \bar{\beta}\, X\, Y_{tot}$$

The evolutionary dynamics, the dynamics of the frequency distribution $\phi(g,e)$, is given by:

$$\frac{d\phi(g,e)}{dt} = \frac{1}{Y_{tot}^2} \left( \frac{dY(g,e)}{dt} Y_{tot} - Y(g,e) \frac{dY_{tot}}{dt} \right)$$
$$= X \left( \int\limits_{\gamma=-\infty}^{\infty} \int\limits_{\epsilon=-\infty}^{\infty} \beta(\gamma+\epsilon)\, \phi(\gamma,\epsilon) P(e) Q(\gamma \to g) d\epsilon\, d\gamma - \bar{\beta}\, \phi(g,e) \right)$$
$$+ (\bar{\mu} - \mu(g+e))\, \phi(g,e)$$

It follows that the mean environmental effect in the population evolves as:

$$\frac{d\bar{e}}{dt} = \int_{g=-\infty}^{\infty} \int_{e=-\infty}^{\infty} e\,\frac{d\phi(g,e)}{dt}de\,dg$$

$$= X\left[\int_{\gamma=-\infty}^{\infty}\int_{\epsilon=-\infty}^{\infty}\beta(\gamma+\epsilon)\,\phi(\gamma,\epsilon)\left(\int_{e=-\infty}^{\infty}e\,P(e)de\right)\left(\int_{g=-\infty}^{\infty}Q(\gamma\to g)dg\right)d\epsilon\,d\gamma - \int_{g=-\infty}^{\infty}\int_{e=-\infty}^{\infty}e\,\bar{\beta}\,\phi(g,e)de\,dg\right]$$

$$+\int_{g=-\infty}^{\infty}\int_{e=-\infty}^{\infty}e\,(\bar{\mu}-\mu(g+e))\,\phi(g,e)de\,dg$$

$$= -X\,\bar{\beta}\,\bar{e}+\bar{e}\bar{\mu}-\int_{g=-\infty}^{\infty}\int_{e=-\infty}^{\infty}e\,\mu(g+e)\,\phi(g,e)de\,dg = -X\,\bar{\beta}\,\bar{e}-\mathrm{cov}[\mu,e]$$

Transmission acts as a force that brings the environmental effect to zero, as the mean of the environmental effect in new recipients is 0. However the positive correlation between the death rate and the environmental effect tend to decrease the environmental effect, as those individuals with higher environmental effect will die faster.

The mean genetic effect in the population evolves as:

$$\frac{d\bar{g}}{dt} = \int_{g=-\infty}^{\infty}\int_{e=-\infty}^{\infty}g\,\frac{d\phi(g,e)}{dt}de\,dg$$

$$= X\left(\int_{\gamma=-\infty}^{\infty}\int_{\epsilon=-\infty}^{\infty}\beta(\gamma+\epsilon)\,\phi(\gamma,\epsilon)\left(\int_{g=-\infty}^{\infty}g\,Q(\gamma\to g)dg\right)d\epsilon\,d\gamma-\bar{\beta}\,\bar{g}\right)$$

$$+\left(\bar{\mu}\,\bar{g}-\int_{g=-\infty}^{\infty}\int_{e=-\infty}^{\infty}g\,\mu(g+e)\,\phi(g,e)de\,dg\right)$$

As $\int_{g=-\infty}^{\infty}g\,Q(\gamma\to g)dg = \gamma+\alpha$, this simplifies into:

$$\frac{d\bar{g}}{dt} = X\left(\int_{\gamma=-\infty}^{\infty}\int_{\epsilon=-\infty}^{\infty}\beta(\gamma+\epsilon)\,\phi(\gamma,\epsilon)\,\gamma\,d\epsilon\,d\gamma+\alpha\bar{\beta}-\bar{\beta}\,\bar{g}\right)$$

$$+\left(\bar{\mu}\,\bar{g}-\int_{g=-\infty}^{\infty}\int_{e=-\infty}^{\infty}g\,\mu(g+e)\,\phi(g,e)de\,dg\right)$$

$$= X\,\mathrm{cov}[\beta,g]+\alpha\bar{\beta}X-\mathrm{cov}[\mu,g]$$

$$\frac{d\bar{g}}{dt} = \mathrm{cov}[\beta X-\mu,g]+\alpha\bar{\beta}X$$

The first term in the equation above is the Robertson-Price identity, as it expresses the change in the genetic value of the trait as the covariance between $\beta X-\mu$, a measure of fitness, and the genetic value. The second term $\alpha\bar{\beta}X$ represents the directional effect of biased mutations on virulence evolution, which is proportional to incidence. The mean viral load evolves as:

$$\frac{d\bar{v}}{dt} = X\,\mathrm{cov}[\beta,g]+\alpha\bar{\beta}X-\mathrm{cov}[\mu,g]-X\,\bar{\beta}\,\bar{e}-\mathrm{cov}[\mu,e]$$

$$= X\,(\mathrm{cov}[\beta,g]+\alpha\bar{\beta}-\bar{\beta}\,\bar{e})-\mathrm{cov}[\mu,v]$$

The evolution of the mean viral load depends on the covariance between transmission and the genetic value of viral load. That's because upon transmission, only the genetic value is faithfully transmitted to the recipient. In contrast, the evolution of the viral load depends on the covariance between the death rate and the full viral load. This is because even if the viral load was fully determined by environmental effect, those individuals with higher environmental effect will die faster and therefore the mean viral load will tend to decrease in the population. The evolution of the mean viral load depends, in general, on the number of susceptible individuals, which is itself changing through time. When there are a large number of susceptible individuals in the population, the first term may dominate the

equation and higher virulence evolves (**Lenski and May, 1994**; **Shirreff et al., 2011**; **Berngruber et al., 2013**).

To understand further how the genetic component of viral load evolve, we develop an approximation inspired by a classical quantitative genetics result (**Lande, 1976**). We will demonstrate below our main result: the change in genetic component of SPVL due to selection can be approximated, if SPVL follows a normal distribution, the $g$ and $e$ components are independent, and the population is at demographic equilibrium, by:

$$\text{cov}[\beta X - \mu, g] = V_G \frac{\bar{\mu}^2}{\bar{\beta}} \frac{\partial(\bar{\beta}/\bar{\mu})}{\partial \bar{g}}$$

This equation implies that, in the absence of biased mutation, the mean genetic value will evolve at a rate proportional to the additive genetic variance $V_G$, climbing the fitness function $\frac{\partial(\bar{\beta}/\bar{\mu})}{\partial \bar{g}}$ until it reaches the value maximizing $\bar{\beta}/\bar{\mu}$.

To demonstrate this relationship, we write the derivative of $\bar{\beta}/\bar{\mu}$ with respect to $\bar{g}$.

$$\frac{\partial(\bar{\beta}/\bar{\mu})}{\partial \bar{g}} = \frac{\partial}{\partial \bar{g}} \left( \frac{\int_{g=-\infty}^{\infty} \int_{e=-\infty}^{\infty} \beta(g+e)\, \phi(g,e) dg\, de}{\int_{g=-\infty}^{\infty} \int_{e=-\infty}^{\infty} \mu(g+e)\, \phi(g,e) dg\, de} \right)$$

$$= \frac{1}{\bar{\mu}^2} \left( \bar{\mu} \int_{g=-\infty}^{\infty} \int_{e=-\infty}^{\infty} \beta(g+e)\, \frac{\partial \phi(g,e)}{\partial \bar{g}} dg\, de - \bar{\beta} \int_{g=-\infty}^{\infty} \int_{e=-\infty}^{\infty} \mu(g+e)\, \frac{\partial \phi(g,e)}{\partial \bar{g}} dg\, de \right)$$

Assuming $\phi(g,e)$ is the density of a normal distribution with means $\bar{g}$ and $\bar{e}$, with variances $V_G$ and $V_E$, and neglecting any covariance that might arise between $g$ and $e$ in the population, we have:

$$\frac{\partial \phi(g,e)}{\partial \bar{g}} = \frac{1}{V_G} \phi(g,e)(g - \bar{g})$$

Replacing yields:

$$\frac{\partial(\bar{\beta}/\bar{\mu})}{\partial \bar{g}} = \frac{1}{\bar{\mu}^2} \left( \bar{\mu} \int_{g=-\infty}^{\infty} \int_{e=-\infty}^{\infty} \beta(g+e) \left( \frac{1}{V_G} \phi(g,e)(g-\bar{g}) \right) dg\, de - \bar{\beta} \int_{g=-\infty}^{\infty} \int_{e=-\infty}^{\infty} \mu(g+e) \left( \frac{1}{V_G} \phi(g,e)(g-\bar{g}) \right) dg\, de \right)$$

And rearranging:

$$\frac{\partial(\bar{\beta}/\bar{\mu})}{\partial \bar{g}} = \frac{1}{V_G} \frac{\bar{\beta}}{\bar{\mu}^2} \left( \frac{\bar{\mu}}{\bar{\beta}} \text{cov}[\beta, g] - \text{cov}[\mu, g] \right)$$

When the number of susceptible individuals has settled to its equilibrium value $X^* = \bar{\mu}/\bar{\beta}$, we have $\frac{\bar{\mu}}{\bar{\beta}} \text{cov}[\beta, g] - \text{cov}[\mu, g] = X^* \text{cov}[\beta, g] - \text{cov}[\mu, g] = \text{cov}[\beta X^* - \mu, g] = d\bar{g}/dt$. Thus we obtain:

$$\frac{\partial(\bar{\beta}/\bar{\mu})}{\partial \bar{g}} = \frac{1}{V_G} \frac{\bar{\beta}}{\bar{\mu}^2} \frac{d\bar{g}}{dt}$$

Re-arranging yields the result presented above. All in all, the change in genetic value due to selection and biased mutation, at demographic equilibrium ($X^* = \bar{\mu}/\bar{\beta}$), can be rewritten as:

$$\frac{d\bar{g}}{dt} = V_G \frac{\bar{\mu}^2}{\bar{\beta}} \frac{\partial(\bar{\beta}/\bar{\mu})}{\partial\bar{g}} + \alpha\bar{\mu}$$

Note that a similar equation can be found for $\mathrm{cov}[\beta X - \mu, g]$ by Taylor-expanding $\beta X - \mu$ in $g$ around $\bar{g}$, holding $X$ constant.

The analytical prediction concerns mean genetic effect of SPVL in *prevalent* cases. Mean genetic effect of SPVL in *incident* cases would be obtained by weighting the distribution of SPVL in prevalent cases by the rate of transmission, and adding up the effect of within-host evolution on SPVL. Simulations show the rate of evolution in incident cases is similar to that in prevalent cases (**Figure 2—figure supplement 6**).

## Parameterization of the model

### Main model

We simulated the differential equation that governs the evolution of the distribution of genetic and environmental effects in the population $Y(g, e)$, and the number of susceptible individuals, from year 1995 to 2015. We discretized the space of possible $(g, e)$ into bins of size 0.1 $\log_{10}$ copies/mL.

To parameterize the model, we used the best-fit relationships for transmission $\beta(v)$ and the severity of infection $\mu(v)$ as inferred from the data. The severity of infection is inversely related to the time to AIDS, such that $\mu(v) = 1/t_{AIDS}(v)$ where $t_{AIDS}(v)$ is the expected time to AIDS for viral load *v*. The transmission rates in the population are assumed to be proportional to the transmission rates fitted in serodiscordant couples. The constant of proportionality, which we call the 'baseline' transmission rate $\beta_0 = 5. \ 10^{-8}$, was estimated to give a realistic equilibrium prevalence of 14%, corresponding to the average prevalence in the 1995–2013 period across communities in Rakai. Indeed, prevalence was constant in time in the communities surveyed here (prevalence ranges from 12 to 29% across communities). Note that in other communities or in Uganda as a whole there is indication that prevalence was maximum around 1992 and declined since then (**Stoneburner et al., 1996**; **Stoneburner and Low-Beer, 2004**; **Yebra et al., 2015**).

Similarly, the birth rate $b = 0.0178$ per year was chosen such that the stable population size remains around 20 M.

The parameters that govern the inheritance of SPVL were $\sigma_{mut} = 0.15$ and $\sigma_e = 0.76$. Using these values, heritability remained constant at around 36%, corresponding to the best estimate of heritability in this cohort (**Hollingsworth et al., 2010**), and phenotypic variance, the variance of SPVL in the population, remained constant at around $V_P = 0.91$ as in the data. The initial variances of environmental and genetic components were set at $(1 - h^2)V_P$ and $h^2 V_P$. The initial average SPVL was chosen such that the SPVL among incident cases was at 4.72 $\log_{10}$ copies/mL as in our dataset. Specifically, 4.72 is the value predicted by the linear model that explains SPVL as a function of gender, age, subtype, and date of seroconversion, when date is year 1995, the sex ratio is 0.5 and the proportion of the different subtypes is that observed in 1995. Under this linear model, the predicted SPVL at year 2015 is 4.23 $\log_{10}$ copies/mL.

### Biased mutation

In order to model the fact that mutations that decrease SPVL may be more frequent than those that increase SPVL, we assumed that $Q(\gamma \rightarrow g)$, the probability that the recipient has genetic value $g$ given that the donor has genetic value $\gamma$, is given by the density of a normal distribution with a non-zero mean $\alpha$, and variance $\sigma_{mut}^2$, evaluated at $g - \gamma$. We explored three scenarios for the plausible mean mutational effect.

## Subtype-specific model

Simulations were conducted as for the main model. Here the 'baseline' transmission rate was $\beta_0 = 3.2\ 10^{-8}$ (estimated to give a realistic equilibrium prevalence of 14%). The birth rate was $b = 0.0181$ per year. We used scenario 2 for the impact of within-host evolution, i.e. $\alpha = -0.093\ \log_{10}$ copies/mL. The average SPVL in 1995 for subtype A, D and R was set to 4.50, 4.80, and 4.61, such that the average SPVL in incident cases was 4.58, 4.79, and 4.66 $\log_{10}$copies per mL of blood. These are the SPVL values predicted for each subtype by the linear model for SPVL as a function of gender, age, subtype, and date of seroconversion, when this date is 1995, and the sex ratio is 0.5.

The frequencies of the three types in 1995 were $p_A$=0.12, $p_D$=0.82, $p_R$=0.06, such that the frequencies of the three types in incident cases was $p_A$=0.17, $p_D$=0.7, $p_R$=0.13, as predicted by a multinomial linear model fitting the frequency of the three subtypes in the data as a function of seroconversion date.

