## [Decision Letter]

Thank you for submitting your article "A transmission-virulence evolutionary trade-off drives attenuation of HIV-1 in Uganda" for consideration by *eLife*. Your article has been favorably evaluated by Prabhat Jha as the Senior Editor and three reviewers, including Richard A Neher (Reviewer #1), who is a member of our Board of Reviewing Editors, and Troy Day (Reviewer #3).

The reviewers have discussed the reviews with one another and the Reviewing Editor has drafted this decision to help you prepare a revised submission.

Summary:

In this manuscript, you present data on HIV transmission and set point viral load (SPVL) evolution and show that these data are consistent with a model of evolution in which SPVL is driven to a value maximizing transmission. This value corresponds to an optimal trade-off between high per-contact transmission probability, favoring high SPVL, and long periods of asymptomatic infection to allow for many potential transmission events (favoring low SPVL). The data from a large cohort of sero-discordant couples in Uganda followed for 20 years are well suited to parameterize the model and test its hypotheses. All reviewers agreed that you present much needed empirical support for a popular evolutionary hypothesis. However, in the reviews and ensuing discussion, we identified a number of points that need to be addressed.

Essential revisions:

1) Definition of virulence:

In most of the theoretical literature virulence is measured as the mortality rate that a pathogen induces upon its host during infection. And this is how you model it (subsection “Epidemiological and evolutionary modelling”, equation numbers would have helped) – an additional mortality rate that depends on the strain type. However, in HIV different strains don't induce different mortality rates during the asymptomatic phase, but more "virulent" strains (those with higher SPVL) shorten the asymptomatic phase. An extended discussion of the trade-off (Introduction, first paragraph) and the relationship of virulence, disease progression, and the expected number of transmission would help many readers. Are there quantitative differences depending on whether virulence variation is modeled as mortality variation or as variation in disease progression? Does the individual based model account for the "time to AIDS"-distribution?

2) Intra-host SPVL trends:

The assumption that mutations that accumulate during chronic infection are randomly drawn from inferred or measured distributions fitness effects is not appropriate. Most deleterious mutations will be pruned by purifying selection. During within-host evolution, mutations will fix that allow the virus to evade host immune selection or increase viral fitness, for example via reversions and compensatory mutations. The authors need to demonstrate robustness of their results to the choice of α and the variance of *Q(γ->g)*.

3) Within vs. between subtype dynamics:

SPVL differs between subtypes and the authors show that relative subtype frequencies change as expected from the virulence/transmission trade-off. We would like to see a discussion of what part of the observed SPVL decline is due to a shift in subtype frequencies and what part is due to intra-subtype evolution.

4) Confounding by ART:

Unreported ART after 2004 remains a concern. In –the eleventh paragraph of the Results, the authors state that the same trend is observed in early years, but that the results are non-significant. You also state that "[…] after the non-significant [...] factors were removed". Does this mean that you did *not* find similar trends when those variables are included? Why are there way fewer data points after 2004? Especially extremely high viral load measurements seem to be missing from the later years. Could the roll-out of ART have led to a reduced representation of rapid progressors in the data set since they were treated early?

5) The study shows that an evolutionary model produces temporal patterns that are similar to the temporal patterns in the data. This match is *consistent* with evolution explaining the data but that is quite different from such a match *suggesting* that the evolutionary explanation is correct. Such claims (Results, tenth paragraph and Discussion, third paragraph) should be toned down.

6) Equilibrium assumption for *X*:

The authors assume the fraction of susceptible individuals *X* is at equilibrium μ¯*/*g¯. How robust are the results to this assumption? It might be interesting to note that one arrives at similar expression for mean SPVL evolution by Taylor-expanding *βX –*μ in *g* while assuming *X* is constant.

---

## [Author Response]

*[…] Essential revisions:*

*1) Definition of virulence:*

*In most of the theoretical literature virulence is measured as the mortality rate that a pathogen induces upon its host during infection. And this is how you model it (subsection “Epidemiological and evolutionary modelling”, equation numbers would have helped) – an additional mortality rate that depends on the strain type. However, in HIV different strains don't induce different mortality rates during the asymptomatic phase, but more "virulent" strains (those with higher SPVL) shorten the asymptomatic phase. An extended discussion of the trade-off (Introduction, first paragraph) and the relationship of virulence, disease progression, and the expected number of transmission would help many readers. Are there quantitative differences depending on whether virulence variation is modeled as mortality variation or as variation in disease progression? Does the individual based model account for the "time to AIDS"-distribution?*

We have added clarifications of the relationship between the inferred gamma distributed time to AIDS and the constant death rate assumed in the ODE model in several places in the text.

Assuming little transmission occurs between the onset of AIDS and death, from the parasite point of view the onset of AIDS and death are effectively equivalent (Introduction, first paragraph).

We fitted a gamma-distributed time to AIDS to the data while in the ODE model we used a constant rate of death with the same expected time to AIDS. This is now made precise in the fifth paragraph of the Results. The constant death rate used in ODE model is equivalent to an exponentially distributed time to AIDS. Whether time to AIDS is gamma distributed or exponentially distributed does not make a large difference for SPVL evolution: (i) the inferred gamma distribution is not very different from an exponential in our dataset, as the shape parameter is 1.2 (it would be 1 in an exponential distribution) (Results, fourth paragraph). (ii) the individual based model does account for the gamma distribution of time to AIDS and the SPVL evolution is not very different in this model. Specifically in the IBM we explicitly described disease progression as several discrete CD4 count categories, with a transition rate between them. These transition rates were tuned to reproduce the inferred gamma distribution of time to AIDS (Results, eighth paragraph).

We hope these explanations clarify this issue.

*2) Intra-host SPVL trends:*

*The assumption that mutations that accumulate during chronic infection are randomly drawn from inferred or measured distributions fitness effects is not appropriate. Most deleterious mutations will be pruned by purifying selection. During within-host evolution, mutations will fix that allow the virus to evade host immune selection or increase viral fitness, for example via reversions and compensatory mutations. The authors need to demonstrate robustness of their results to the choice of α and the variance of Q(γ->g).*

This is an important point.

First of all, the *variance* of the within-host effect of mutation was parameterized to reproduce the observed phenotypic variance and heritability of 36% measured in the Rakai cohort (subsection “Epidemiological and evolutionary modelling”, seventh paragraph). There is now good evidence that heritability of SPVL is around 30% (reviewed in Fraser et al. Science 2014; see also Mitov and Stadler bioRxiv 2016), so we have good confidence in this parameter.

However, there is little data on the mean effect of mutations arising during within-host evolution on SPVL. From the Price equation, the overall evolution of SPVL depends on the balance between between-host evolution (under the virulence-transmission tradeoff) and within-host evolution. Clearly, a strong effect of within-host evolution will swamp adaptation to the virulence-transmission tradeoff.

We now discuss more fully the type of mutations that may evolve within the host and how this might affect SPVL (Results, seventh paragraph and subsection “Epidemiological and evolutionary modelling”, seventh paragraph). Many different types of mutations may evolve within the host, and little is known on the net effect of these processes on SPVL. Many of the evolving mutations, for example CTL escape mutations, are beneficial conditional on the host genotype, and allow immune escape but carry a cost in terms of replicative capacity (RC). Moreover, unconditionally beneficial mutations, for example mutations increasing RC, may evolve, and it is also plausible that weakly deleterious mutations fix due to random drift.

We reviewed the literature and explored three scenarios representing a plausible range of effects of within-host evolution. The first two scenarios represent an increase in frequency of immune escape mutations, with a strong RC cost (scenario 1) or a moderate RC cost (scenario 2). Scenario 1 is parameterized based on the inferred decline in SPVL in Botswana, explained by viral adaptation to the host HLA make-up (Payne et al. PNAS 2012). Scenario 2 is parameterized assuming the RC cost of escape mutations is similar to that of random mutations. The third scenario considers the evolution of mutations increasing RC within the host, parameterized using the only study that inferred such an effect (Kouyos et al. PLOS Pathogens 2011). See subsection “Epidemiological and evolutionary modelling”, seventh paragraph for details of these scenarios.

In all these scenarios, the average SPVL is declining and the component of SPVL evolution due to adaptation to the virulence-transmission tradeoff is of similar magnitude compared to within-host evolution. In spite of uncertainty on the sign and magnitude of the within-host evolution effect, the important point is that the virulence-transmission tradeoff predicts a decline in SPVL, that this decline is qualitatively observed in the data, and that the virulence-transmission force accounts for ~ 50% of this decline.

*3) Within vs. between subtype dynamics:*

*SPVL differs between subtypes and the authors show that relative subtype frequencies change as expected from the virulence/transmission trade-off. We would like to see a discussion of what part of the observed SPVL decline is due to a shift in subtype frequencies and what part is due to intra-subtype evolution.*

We decomposed the total change in average SPVL into the sum of two components, one due to changes in subtype frequency, one due to within-subtype changes in SPVL (Materials and methods subsection “Contributions of within-subtype and between-subtype evolution to SPVL trends”). While the within-subtype change is -0.022 log_10_ copies/mL/year, the additional change due to changes in subtype frequency is much smaller, at -0.003 log_10_ copies/mL/year Most of the change in average SPVL is thus due to within-subtype evolution. We added this result to the last paragraph of the Results section.

*4) Confounding by ART:*

*Unreported ART after 2004 remains a concern. In –the eleventh paragraph of the Results, the authors state that the same trend is observed in early years, but that the results are non-significant. You also state that "[…] after the non-significant […] factors were removed". Does this mean that you did not find similar trends when those variables are included?*

In the pre-2004 subset of data, all SPVL but one were measured with the Roche 1.5 assay, so we did not consider a potential impact of the assay. We had little power to distinguish between “laboratory” and “calendar time” effects because of a strong correlation between these factors (∆AIC = -1.9 for a model with “laboratory” relative to a model with “calendar time”). However, we know from the analysis of the full dataset that “laboratory” has no significant effect on SPVL, and furthermore the inferred effects of “laboratory” in the pre-2004 subset are consistent with confounding by calendar time and different from those of the full dataset, which suggests the temporal effect is the genuine effect here. We explain this in more detail in the eleventh paragraph of the Results.

ART causes a clear decline in viremia, which most of the time goes below the detection limit. We do not see this pattern in post-2004 participants for whom we do not have ART information, suggesting they are not under ART. We added [Supplementary-material SD3-data] showing the individual viral load trajectories in 603 participants who have a SPVL value. This is a bit cumbersome, but allows clear visualization of the way we compute SPVL and verification that viral load trajectories in post-2004 participants without ART information do not present patterns suggestive of unreported ART (Results, eleventh paragraph).

Moreover, the downward trend in average SPVL is not caused by an excess of low SPVL values at more recent time points. The entire distribution of SPVL is shifted downward as shown by the new Figure 2—figure supplement 4 (Results, eleventh paragraph).

*Why are there way fewer data points after 2004?*

The number of points per time period is contingent on the studies that were running in the Rakai cohort.

A molecular epidemiology study conducted by the Walter-Reed Institute ran in the late 1990s- early 2000s. As part of this study all seroconverters were enrolled and viral loads were completed regardless of time since seroconversion. There were also a few other viral load studies conducted the early 1990s.

After 2004, viral loads have been done for some sub-studies of the RCCS but largely as part of routine care. This means that people who did not enter care or were not yet eligible or who were seen at non-RCCS clinics would probably not have a viral load. One could hypothesize that sicker people are more likely to seek care. In this scenario, a disproportionately higher number of individuals with high viral loads would be tested post 2004 but if anything that would bias results away from the downward trend.

*Especially extremely high viral load measurements seem to be missing from the later years. Could the roll-out of ART have led to a reduced representation of rapid progressors in the data set since they were treated early?*

The decline in SPVL is not due to high viral loads missing in recent time points. The entire distribution of SPVL is shifted downward as shown by the new Figure 2—figure supplement 4.

*5) The study shows that an evolutionary model produces temporal patterns that are similar to the temporal patterns in the data. This match is consistent with evolution explaining the data but that is quite different from such a match suggesting that the evolutionary explanation is correct. Such claims (Results, tenth paragraph and Discussion, third paragraph) should be toned down.*

We have toned down these claims in several places including the title, and Results, ninth paragraph and Discussion, first paragraph.

*6) Equilibrium assumption for X:*

*The authors assume the fraction of susceptible individuals X is at equilibrium*
μ¯*/*g¯*. How robust are the results to this assumption? It might be interesting to note that one arrives at similar expression for mean SPVL evolution by Taylor-expanding βX –*
μ
*in g while assuming X is constant.*

We emphasise that prevalence is approximately constant in the Rakai communities (Figure 2—figure supplement 1). A hypothetical reduction in prevalence would tend to accelerate the decline in average SPVL as predicted by the analysis, but the effect is weak. We now show an example of this phenomenon on Figure 2—figure supplement 5, in a case where prevalence declines from 20% to 5% over 20 years.

We added in the supplement the note on the Taylor expansion leading to a similar result. Thank you for this suggestion.